# Neural Solver Selection for Combinatorial Optimization

## Abstract

Machine learning has increasingly been employed to solve NP-hard combinatorial optimization problems, resulting in the emergence of neural solvers that demonstrate remarkable performance, even with minimal domain-specific knowledge. To date, the community has created numerous open-source neural solvers with distinct motivations and inductive biases. While considerable efforts are devoted to designing powerful single solvers, our findings reveal that existing solvers typically demonstrate complementary performance across different problem instances. This suggests that significant improvements could be achieved through effective coordination of neural solvers at the instance level. In this work, we propose the first general framework to coordinate the neural solvers, which involves feature extraction, selection model, and selection strategy, aiming to allocate each instance to the most suitable solvers. To instantiate, we collect several typical neural solvers with state-of-the-art performance as alternatives, and explore various methods for each component of the framework. We evaluated our framework on two extensively studied combinatorial optimization problems, Traveling Salesman Problem (TSP) and Capacitated Vehicle Routing Problem (CVRP). Experimental results show that the proposed framework can effectively distribute instances and the resulting composite solver can achieve significantly better performance (e.g., reduce the optimality gap by 0.88% on TSPLIB and 0.71% on CVRPLIB) than the best individual neural solver with little extra time cost.

## 1 Introduction

Combinatorial Optimization Problems (COPs) involve finding an optimal solution over a set of combinatorial alternatives, which has broad and important applications such as logistics (Konstantakopoulos et al., 2022) and manufacturing (Zhang et al., 2019). To solve COPs, traditional approaches usually depend on heuristics designed by experts, requiring extensive domain knowledge and considerable effort. Recently, machine learning techniques have been introduced to automatically discover effective heuristics for COPs (Bengio et al., 2021; Cappart et al., 2023), leading to the burgeoning development of end-to-end neural solvers that employ deep neural networks to generate solutions for problem instances (Bello et al., 2017; Kool et al., 2019; Joshi et al., 2019). Compared to traditional approaches, these end-to-end neural solvers can not only get rid of the heavy reliance on expertise, but also realize better inference efficiency (Bello et al., 2017).

To enhance the capabilities of neural solvers, a variety of methods have been proposed, with intensive effort on the design of frameworks, network architectures, and training procedures. For example, to improve the performance across different distributions, Jiang et al. (2022) proposed adaptively joint training over varied distributions, and Bi et al. (2022) leveraged knowledge distillation to integrate the models trained on different distributions. For generalization on large-scale instances, Fu et al. (2021) implemented a divide-and-conquer strategy, Luo et al. (2023) proposed a heavy-decoder structure to better capture the relationship among nodes, while Gao et al. (2024) utilized the local transferability and introduced an additional local policy model. Diffusion models (Sun & Yang, 2023) have also been adapted to generate the distribution of optimal solutions, demonstrating impressive results. More works include bisimulation quotienting (Drakulic et al., 2023), latent space search (Chalumeau et al., 2023), local reconstruction (Cheng et al., 2023; Ye et al., 2024; Zheng et al., 2024) and so on.

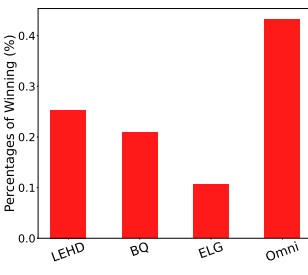
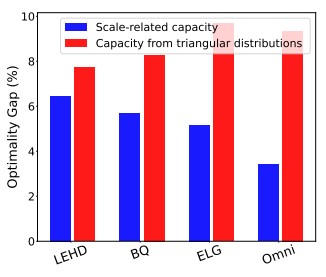
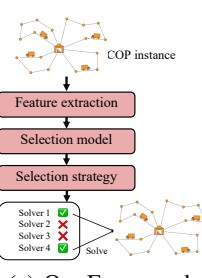

(a) Percentages of Winning      (b) Optimality Gap (Average)      (c) Our Framework

Figure 1: (a), (b): Observation from the comparison of prevailing neural solvers at instance level. Details of the settings are provided in Section 4.1. (c): Our proposed coordination framework.

As various neural solvers are emerging in the community, the state-of-the-art records for the overall performance on benchmark problems are frequently refreshed. However, the detailed comparison of these neural solvers on each instance has been rarely discussed. Here, we empirically examined the performance of several prevailing neural solvers on each instance, as illustrated in Figure 1(a) and 1(b). As expected from the no-free-lunch theorem (Wolpert & Macreday, 1997), we find that

- As shown in Figure 1(a), there exists no single neural solver that can dominate all the other neural solvers on every instance, and different neural solvers win on different instances, demonstrating their complementary performance at instance level.
- As shown in Figure 1(b), the modification of instance distribution can almost reverse the domination relationship of neural solvers, which further verifies that different neural solvers are good at instances with specific characteristics due to their intrinsic inductive biases.

These observations suggest that it may potentially bring impressive improvements to the overall performance, if multiple neural solvers are coordinated to solve instances together. In fact, recent works have already made preliminary attempts from the perspective of ensemble learning (Jiang et al., 2023) and population-based training (Grinsztajn et al., 2023). However, their individual solvers share the same architecture, resulting in limited diversity. On the other hand, as all of the individual solvers should run during inference, these methods can hardly achieve ideal efficiency.

Motivated by the observations above, we, for the first time, propose a general framework to coordinate end-to-end neural solvers for COPs at instance level by selecting suitable individual solvers for each instance, as illustrated in Figure 1(c). Specifically, our proposed framework consists of three key components, which are summarized as follows:

- **Feature extraction:** For each problem instance, extract features that can be effectively used to identify their characteristics.
- **Selection model:** Based on the features of instances, train a selection model that can be utilized to identify suitable solvers for each instance.
- **Selection strategy:** Due to the intricate structures of COPs, using only the most suitable individual solver predicted by the selection model may fail. Therefore, it is important to design robust selection strategies based on the confidence of the selection model.

To verify the effectiveness of our proposed framework, we collect several prevailing open-source neural solvers and their released models with competitive performance in the community to construct the pool of individual solvers, and provide several implementations for each component of the framework. For feature extraction, we utilize the graph attention network (Veličković et al., 2018; Kool et al., 2019) to encode COP instances, and further propose a refined encoder with pooling to leverage the hierarchical structures of COPs. For selection model, we train it from the perspective of classification and ranking, respectively. We also implement several selection strategies, including top-$k$ selection, rejection-based selection, and so on. Detailed descriptions are provided in Section 3. We conduct experiments on two widely studied COPs: Traveling Salesman Problem (TSP) and Capacitated Vehicle Routing Problem (CVRP). Experimental results exhibit that our framework can generally select suitable individual solvers for each instance to achieve significantly better per-

formance with limited extra time consumption. Compared to the best individual solver, our framework reduces the optimality gap by 0.82% on synthetic TSP, 2.00% on synthetic CVRP, 0.88% on TSPLIB (Reinelt, 1991), and 0.71% on CVRPLIB Set-X (Uchoa et al., 2017). As this is the first preliminary attempt on neural solver selection for COPs, we also analyze the influence of various implementations of components, and provide some discussion on future improvements.

## 2 RELATED WORKS

### 2.1 END-TO-END NEURAL SOLVERS FOR COPs

Traditional approaches for COPs have achieved impressive results, but they often rely on problem-specific heuristics and domain knowledge by experts (Helsgaun, 2000; 2017). Instead, recent efforts focus on utilizing end-to-end learning methods. A prominent fashion is autoregression, which employs graph neural networks in an encoder-decoder framework and progressively extends a partial solution until a complete solution is constructed (Vinyals et al., 2015; Bello et al., 2017; Kool et al., 2019). However, these methods tend to exhibit poor generalization performance across distributions and scales (Joshi et al., 2022). To address the generalization issue, considerable efforts have been dedicated within the community. For example, Zhou et al. (2023) took various distributions and scales as different learning tasks and adopted meta-learning over them to obtain a generalizable model. Bi et al. (2022) leveraged knowledge distillation, where models trained on different distributions are utilized as teacher models for one generalizable student model. Liu et al. (2024) used the idea of prompt learning to realize zero-shot adaptation of the pertained model by selecting the most matched prompt for instances. More efforts in autoregressive methods include instance-conditioned adaptation (Zhou et al., 2024a), adversarial training (Wang et al., 2024) and nested local views (Fang et al., 2024), to name a few.

Another popular kind of end-to-end learning methods is non-regressive, which predicts or generates the distributions of potential solutions. Typically, Joshi et al. (2019); Ye et al. (2023) employed graph neural networks to predict the probability of components appearing in an optimal solution, represented with the form of heatmap. Diffusion models (Sun & Yang, 2023; Sanokowski et al., 2024) have also been adapted to generate the distribution of optimal solutions, demonstrating better expressiveness than classical push-forward generative models (Salmona et al., 2022).

### 2.2 SOLVING COPs WITH MULTIPLE NEURAL SOLVERS

Recent studies have made preliminary attempts to integrate multiple neural solvers to enhance overall performance on COPs. For example, Jiang et al. (2023) adopted ensemble learning, where multiple neural solvers with identical architecture are trained on different instance distributions through Bootstrap sampling to ensure diversity. During inference, the outputs of all the solvers are gathered by average at each action step. Grinsztajn et al. (2023) proposed a population-based training method Poppy, where multiple decoders with a shared encoder are trained simultaneously as a population of solvers, with a reward targeting at maximizing the overall performance of the population. When solving a problem instance, each solver generates solutions independently, and the best solution is selected as the final result. However, these works suffer from heavy computation cost as multiple solvers have to be run for each instance. Even they propose to share a common encoder for each solver, experimental results still demonstrate undesired inference time (Grinsztajn et al., 2023). On the other hand, different solvers share the same neural architecture, which may limit the diversity and thus the final performance.

Consider that the burgeoning community has proposed many methods from various perspectives, resulting in diverse end-to-end neural solvers with different inductive biases. Properly coordinating these neural solvers can potentially bring a significant improvement on overall performance. Motivated by the observation in Figure 1(a) and 1(b), we propose to select suitable ones from a pool of diverse individual solvers for each instance. Note that similar idea has been utilized in the area of algorithm selection (Kerschke et al., 2019) and model selection (Zhang et al., 2023), but has never been explored in the area of neural combinatorial optimization. By solver selection at instance level, any type of (existing or newly constructed) neural solver can be utilized, and only the selected individual solvers need to be run in inference, thereby maintaining high efficiency.

## 3 THE PROPOSED FRAMEWORK

This section will introduce the proposed framework of coordinating neural solvers for COPs. In general, our target is learning to select the suitable solvers for each problem instance. To address this, our proposed framework comprises three key components:

- **Feature extraction**: To select the most suitable neural solvers for each instance, it is essential to extract the instance features, which is challenging as the COPs are usually intricate. In this work, we first utilize the graph attention encoder (Kool et al., 2019) to encode COP instances, and further propose a refined graph encoder with pooling, which can leverage the hierarchical structures of COPs to obtain better features.

- **Selection model**: We train a neural selection model with the graph encoder to identify the most suitable solvers. Specifically, we implement two loss functions from the perspectives of classification and ranking.

- **Selection strategies**: Due to the complexity of COPs, it may be risky to rely solely on the selection model to consistently identify the most suitable solver. To address this, we propose compromise strategies that allow to allocate multiple solvers (if necessary) to a single instance based on the confidence levels of the selection model, pursuing better performance with limited extra cost.

In the following subsections, we will elaborate the three key components in our framework.

### 3.1 FEATURE EXTRACTION

For feature extraction, it depends on the COP to be solved. Here, we use the two most prevailing problems, TSP and CVPR, in the neural solver community for COPs (Kwon et al., 2020; Luo et al., 2023; Drakulic et al., 2023) as examples, which will also be employed in our experiments. TSP and CVRP involve finding optimal routes over a set of nodes. For TSP, the objective is to find the shortest possible route that visits each node exactly once and returns to the starting node. Each TSP instance consists of nodes distributed in Euclidean space. For CVRP, the goal is to plan routes for multiple vehicles to serve customer nodes with varying demands, starting and ending at a depot node, while minimizing the total travel distance and satisfying vehicle capacity constraints (Dantzig & Ramser, 1959). Both TSP and CVRP instances can be represented as fully connected graphs, where nodes correspond to locations (cities or customers). The graph representation makes them suitable for encoding using Graph Neural Networks (GNNs), which can effectively capture the structural information inherent in these problems (Khalil et al., 2017; Kool et al., 2019). In this paper, we design two types of GNN-based encoders tailored for TSP and CVRP instances as follows.

**Graph attention encoder.** We take the CVRP as an example to describe the computation of the graph encoder. The raw features $x \in \mathbb{R}^{N \times 3}$ of a CVRP instance are a set of nodes $\{(x_i, y_i, m_i)|i \in [N]\}$, where $(x_i, y_i)$ are the node coordinates, $m_i$ is the node demand, $N$ is the number of nodes, and $[N]$ denotes the set $\{1, 2, \ldots, N\}$. First, a linear layer is employed on every node for initial node embeddings, i.e., $H^0 = xW$, where $W \in \mathbb{R}^{3 \times d}$ are the weights and $d$ denotes the embedding dimension. Given initial embeddings, multiple graph attention layers (Veličković et al., 2018; Kool et al., 2019) are applied to iteratively update the node embeddings as $H^l = \text{AttentionLayer}^l(H^{l-1})$, where $l \in [L]$ and $L$ is the number of layers. Since the graphs of TSP and CVRP are both fully connected, the graph attention layer covers every pair of nodes and self-connections, which becomes similar to the self-attention mechanism (Vaswani et al., 2017). Details of the attention layer are provided in Appendix A.1. Finally, the node embeddings output by the last layer are averaged to form the instance representation, as most COP encoders (Khalil et al., 2017; Kool et al., 2019) did.

**Hierarchical graph encoder.** Averaging the final node embeddings may result in sub-optimal instance representations that are too flat to effectively capture the hierarchical structures inherent in COPs (Goh et al., 2024). Inspired by (Lee et al., 2019), we design a hierarchical graph attention encoder to address this limitation, which successively downsamples the graph of an instance using *graph pooling*, and aggregates features from each downsampling level to construct a comprehensive graph representation, as illustrated in Figure 2.

The hierarchical graph encoder contains $L$ blocks. In each block, several graph attention layers are first applied, and then the graph pooling layer selects representative nodes to form a coarsened graph that preserves important features. Consider the $l$-th block, where the number of selected nodes is denoted as $N_l$. To quantify the representativeness of each node for graph pooling, an additional graph attention layer is introduced to compute representative scores. Specifically, this graph attention layer computes score embeddings $H_{\text{score}}^l$ based on the current node embeddings $H^l$, which encode rich information about the graph structure and node features. These score embeddings $H_{\text{score}}^l$ are then mapped to scalar representative scores via linear layer. The complete process is shown as follows:

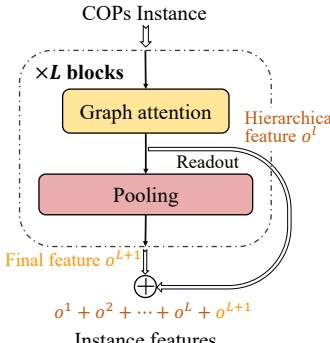

Figure 2: Illustration of the hierarchical graph encoder.

$$H_{\text{score}}^l = \text{AttentionLayer}_{\text{score}}^l(H^l), \quad Z^l = \sigma(H_{\text{score}}^l W_{\text{score}}^l),$$

where $Z^l \in \mathbb{R}^{N^{l-1} \times 1}$ are the representative scores of the $N^{l-1}$ nodes preserved in the $(l-1)$-th block, $\sigma$ is a non-linear function (here we use the $\tanh$ function), and $W_{\text{score}}^l \in \mathbb{R}^{d \times 1}$ are the parameters of the linear layer. Subsequently, we sort the nodes according to their representative scores and select top-$N^l$ nodes ($N^l < N^{l-1} < N$) to preserve. To make this pooling layer trainable via back-propagation (LeCun et al., 2002), we further combine the representative score together with the embeddings of their corresponding nodes as follows: $\tilde{H}^l = H^l + Z^l \mathbf{1}$, where $\mathbf{1} \in \mathbb{R}^{1 \times d}$ is a vector with all the elements being $1$. Intuitively, this operation can move the embeddings of high scored nodes away from the embeddings of low scored nodes.

Figure 2 shows the complete encoder, where $L$ blocks are stacked, and each block is formed by several graph attention layers followed by a pooling layer. By successively applying $L$ encoder blocks, the number of preserved nodes gradually decreases according to $N^l = \alpha \cdot N^{l-1}$ ($\alpha$ is set to 0.8 in our experiments). This process constructs a hierarchy of the original graph and its coarsened versions, enabling the encoder to capture multi-level structural information effectively. Within each block, we apply a *readout* layer that aggregates the embeddings after the graph attention layers by mean pooling and max pooling (Lee et al., 2019), i.e.,

$$\boldsymbol{o}^l = \sigma(\text{Mean}(H^l) \| \text{Max}(H^l)),$$

where $\text{Mean}()$ computes the average embedding over the nodes, $\text{Max}()$ computes the maximum along the column dimension, $\|$ denotes concatenation, and $\sigma$ is a non-linear function. The result $\boldsymbol{o}^l$ provides the representation of the $l$-th coarsened graph. At the last layer, we also readout $\boldsymbol{o}^{L+1}$ from the final embeddings. To form the hierarchical instance representation, we sum the representations of all levels as $\boldsymbol{o} = \sum_{l=1}^{L+1} \boldsymbol{o}^l$.

## 3.2 SELECTION MODEL

We employ a Multiple-Layer Perception (MLP) to predict compatibility scores of neural solvers, where a higher score indicates that it is more suitable to allocate the instance to the corresponding neural solver. This MLP model takes the instance representation and the instance scale $N$ as input and outputs a score vector, where the value of each index is the score of the corresponding neural solver. In summary, the graph encoder and the MLP are cascaded to compose a neural selection model, which can produce the compatibility scores of individual solvers from the raw COP instance in an end-to-end manner. Advanced neural solver features can be incorporated for richer information, as discussed in Section 5. However, we find that even using fixed indices of neural solvers has already been effective, which will be clearly shown in our experiments.

We train the selection model using a supervised dataset comprising thousands of synthetic COP instances. The objective values obtained by the neural solvers are recorded as supervision information. Intuitively, a neural solver with a lower objective value (for minimization) has a higher compatibility score. To learn such desired score output, we employ two losses from the perspectives of classification and ranking.

**Classification.** The selection problem is formulated as a classification task, where the most suitable neural solver for a given instance serves as the ground truth label. By employing classification

loss functions such as cross-entropy loss used in our experiments, we can train a selection model. However, this approach focuses on identifying the optimal neural solver and ignores sub-optimal solvers, which may lead to unsatisfactory performance when the selection is inaccurate.

**Ranking.** The neural solvers can be sorted according to the objective values they obtain, thereby forming a ranking of the given solvers, denoted by $\phi : [M] \to [M]$, where $\phi(i)$ is the index of the rank-$i$ solver and $M$ is the number of solvers. We then train the selection model by maximizing the likelihood of producing correct rankings based on the computed scores (Xia et al., 2008),

$$\max_\theta \mathbb{E}_I[\sum_{i=1}^M \log \frac{\exp(g_\theta(I)_{\phi_I(i)})}{\sum_{j=i}^M \exp(g_\theta(I)_{\phi_I(j)})}],$$

where $g_\theta$ denotes the selection model with parameters $\theta$, $I$ denotes a problem instance, and $\phi_I$ is the ground-truth ranking on instance $I$. This ranking loss can leverage the dominance relationship of all neural solvers, including sub-optimal ones, which thus can make the selection more robust.

### 3.3 SELECTION STRATEGIES

Considering that the intricate structures of COPs may pose great challenge to the selection model, besides greedy selection, we propose several compromise strategies that allow multiple solvers for a single instance based on the confidence level of the selection model, aiming to improve the overall performance with little extra cost.

**Greedy selection.** The most straightforward approach is the greedy selection, which chooses the neural solver with the highest score. This method is efficient since only one solver is executed per instance. However, it may be inaccurate, potentially leading to sub-optimal performance.

**Top-$k$ selection.** The top-$k$ selection method can be adopted for better optimality, where we select and execute the neural solvers with top-$k$ scores for each instance, thus constructing a *portfolio* of multiple solvers. This approach increases the likelihood of including the optimal solver but incurs additional computational overhead due to the execution of multiple solvers.

**Rejection-based selection.** To balance efficiency and effectiveness, we propose the rejection-based selection strategy, which adaptively selects greedy or top-$k$ selection. Recognizing that the confidence of the greedy selection varies across instances, an advanced strategy is to employ the top-$k$ selection for low-confidence instances to enhance performance and utilize only the greedy selection for high-confidence ones to minimize computational cost. To implement this strategy, we can use a confidence measure to determine whether to accept or reject the greedy selection. If the confidence in the greedy selection is below a threshold, we reject it and apply the top-$k$ selection to the instance for improved optimality. In this paper, we adopt the simple yet effective *softmax response* (Hendrycks & Gimpel, 2017) as the confidence measure, and define the threshold by rejecting a certain fraction of test instances with the lowest confidence levels.

**Top-$p$ selection.** We further propose a top-$p$ selection strategy that selects the smallest subset of solvers whose normalized scores (i.e., selection probabilities) sum up to at least $p$. The value of $p$ is predefined or adjusted according to the time budget. Thus, this strategy adaptively determines the number of selected neural solvers by covering a certain amount of probability mass, rather than relying on a fixed number $k$.

## 4 EXPERIMENTS

To examine the effectiveness of our proposed selection framework, we conduct experiments on TSP and CVRP, investigating the following Research Questions (RQ): RQ1: How does the proposed selection framework perform compared to individual neural solvers? RQ2: How does the proposed selection framework perform when the problem distribution shifts and the problem scale increases? RQ3: How do different implementations of components affect the performance of the framework? We introduce the experimental settings in Section 4.1 and investigate the above RQs in Section 4.2. The code and data used in our experiments are provided in the supplementary materials.

## 4.1 EXPERIMENTAL SETTINGS

**Synthetic TSP and CVRP.** We generate synthetic TSP and CVRP instances by sampling node coordinates from Gaussian mixture distributions with randomized covariance matrices. In the case of CVRP, vehicle capacities are generated using either the scale-related capacity or the triangular distribution. We consider varying problem scales, where the scale $N$ is sampled uniformly from $[50, 500]$. More details of the data generation process are provided in Appendix A.2.

**Datasets.** For training, we generate $10,000$ TSP and CVRP instances and apply 8-fold instance augmentation (Kwon et al., 2020). For test, we generate smaller synthetic datasets comprising $1,000$ instances. Figures 1(a) and 1(b) in Section 1 are based on results from the CVRP test dataset. To evaluate the out-of-distribution performance, we utilize two well-known benchmarks with more complex problem distributions and larger problem scales (up to $N = 1002$): TSPLIB (Reinelt, 1991) and CVRPLIB Set-X (Uchoa et al., 2017). For TSPLIB, we select a subset of instances with $N \leq 1002$, and CVRPLIB Set-X includes instances ranging from $N = 100$ to $1000$. These problem scales are larger than the scale $N \in [50, 500]$ of our training datasets.

**Open-source neural solvers.** We choose recent open-source neural solvers with state-of-the-art performance as the candidates, including Omni (Zhou et al., 2023), BQ (Drakulic et al., 2023), LEHD (Luo et al., 2023), DIFUSCO (Sun & Yang, 2023), T2T (Li et al., 2023), ELG (Gao et al., 2024), INViT (Fang et al., 2024) and MVMoE (Zhou et al., 2024b). Greedy decoding is used for all the methods to avoid stochasticity. We set the pomo size to $100$ and the augmentation number to $8$ for the methods based on POMO (Kwon et al., 2020). The number of denoising steps is set to $50$ and the number of 2-opt iterations is set to $100$ for diffusion-based methods. These individual solvers constitute a neural solver zoo. Ideally, if we can always select the best solver from the zoo for each instance, the optimal performance is achieved, which is also the performance upper bound of our selection model. Considering that some neural solvers contribute little to the overall performance, we iteratively eliminate the least contributive solver from the candidates, resulting in a more compact neural solver zoo. This process reduces the zoo size to 7 solvers for TSP and 5 for CVRP. Further details of the elimination procedure are provided in Appendix A.3.

**Hyperparameters.** (1) **Hyperparameters of graph encoders**. For the graph attention encoder, we set the number of layers to $4$. For the hierarchical graph encoder, we use 2 blocks where each block has 2 attention layers. The embedding dimension is set to $128$. Other hyperparameters of encoders can be found in Appendix A.1. (2) **Hyperparameters of training**. The Adam optimizer (Kingma & Ba, 2015) is employed for training, where we set the learning rate to $1 \times 10^{-4}$ and the weight decay to $1 \times 10^{-6}$. The number of epochs is set to $50$. The final model is chosen according to the performance on a validation dataset with $1,000$ synthetic instances. We train 5 selection models using different random seeds and report the mean and standard deviation of their performance. (3) **Hyperparameters of selection strategies**. For the top-$k$ strategy, we set $k = 2$. For the rejection-based strategy, we reject the 20% of instances with the lowest confidence levels (i.e., the highest selection probability of all individual solvers), and apply top-2 selection to these rejected instances. For the top-$p$ strategy, we set $p = 0.5$ for TSP and $p = 0.8$ for CVRP.

**Performance metrics.** Following previous studies, we employ the gap to the best-known solution $\frac{c_I(\hat{\sigma}) - c_I(\sigma^*)}{c_I(\sigma^*)}$ as the performance metric, called optimality gap, where $\hat{\sigma}$ is the solution obtained by each method, $\sigma^*$ is the best-known solution computed by extensive search of expert solvers (Helsgaun, 2017; Vidal, 2022), and $c_I()$ is the cost function of problem instance $I$. We also report the average time to evaluate efficiency, which includes both the running of neural solvers and selection.

## 4.2 EXPERIMENTAL RESULTS

**RQ1: How does the proposed selection framework perform compared to individual neural solvers?** In Table 1, we present the performance of several implementations of our selection framework on synthetic TSP and CVRP, alongside the results of the top-3 individual neural solvers[1]. We can observe that all implementations of our framework outperform the best neural solver on both

---

[1]DIFUSCO and T2T have multiple trained models. We only report the best results of these models.

Table 1: Empirical results on synthetic TSP and CVRP datasets, reporting the mean (standard deviation) over five independent runs. The top three individual solvers are included for comparison, and Oracle denotes the optimal performance for selection, which is computed by running all individual solvers in the zoo for each instance and selecting the best one. The best individual solver and its results are underlined, and the best optimality gaps, excluding Oracle, are highlighted in boldface.

| Methods | TSP | | Methods | CVRP | |
|---|---|---|---|---|---|
| | Gap | Time | | Gap | Time |
| BQ (*3rd*) | 3.00% | 1.40s | LEHD (*3rd*) | 7.37% | 1.01s |
| T2T (*2nd*) | 2.40% | 1.58s | BQ (*2nd*) | 7.20% | 1.59s |
| DIFUSCO (*1st*) | 2.33% | 1.45s | Omni (*1st*) | 6.82% | 0.24s |
| Oracle | 1.24% | 8.93s | Oracle | 4.64% | 4.38s |
| *Selection by classification* | | | *Selection by classification* | | |
| Greedy | 1.94% (0.02%) | 1.36s (0.01s) | Greedy | 5.35% (0.02%) | 0.64s (0.01s) |
| Top-$k$ ($k = 2$) | 1.53% (0.01%) | 2.52s (0.04s) | Top-$k$ ($k = 2$) | **4.81% (0.01%)** | 1.87s (0.03s) |
| Rejection (20%) | 1.81% (0.01%) | 1.63s (0.01s) | Rejection (20%) | 5.19% (0.03%) | 0.77s (0.01s) |
| Top-$p$ ($p = 0.5$) | 1.84% (0.03%) | 1.55s (0.06s) | Top-$p$ ($p = 0.8$) | 5.16% (0.03%) | 0.87s (0.08s) |
| *Selection by ranking* | | | *Selection by ranking* | | |
| Greedy | 1.86% (0.01%) | 1.33s (0.01s) | Greedy | 5.31% (0.01%) | 0.62s (0.01s) |
| Top-$k$ ($k = 2$) | **1.51% (0.02%)** | 2.56s (0.03s) | Top-$k$ ($k = 2$) | 4.82% (0.01%) | 1.90s (0.04s) |
| Rejection (20%) | 1.75% (0.02%) | 1.63s (0.01s) | Rejection (20%) | 5.15% (0.02%) | 0.74s (0.01s) |
| Top-$p$ ($p = 0.5$) | 1.68% (0.02%) | 1.86s (0.07s) | Top-$p$ ($p = 0.8$) | 4.99% (0.02%) | 1.03s (0.03s) |

TSP and CVRP, demonstrating the effectiveness of our framework. For example, using ranking loss and the top-$k$ selection strategy with $k = 2$, our framework achieves average optimality gaps of 1.51% on TSP and 4.82% on CVRP, surpassing the best individual solver's gaps of 2.33% on TSP and 6.82% on CVRP, achieved by DIFUSCO and Omni, respectively. Moreover, except utilizing the top-$k$ strategy, our selection framework is nearly as efficient as running a single solver. In some cases, our framework can obtain better optimality gaps while consuming even less time. For instance, using ranking loss and greedy selection on TSP leads to the average optimality gap 1.86% with 1.33s, while the best individual solver DIFUSCO achieves 2.33% gap with 1.45s. In Table 1, Oracle (the fourth row) denotes the optimal performance for selection, which is obtained by running all individual solvers for each instance and selecting the best one. The best optimality gaps achieved by our selection framework (using ranking loss and top-$k$ selection with $k = 2$) are close to Oracle, with gaps of 1.51% on TSP and 4.81% on CVRP, compared to Oracle's gaps of 1.24% on TSP and 4.64% on CVRP. Furthermore, our framework can offer significant speed advantages over Oracle, e.g., consuming an average time of 2.56s on TSP, whereas Oracle requires an average time of 8.93s. Note that complete results for all individual solvers are provided in Appendix A.10.

**Extension of RQ1: Is the performance of the top-$k$ selection better than the solver portfolio with the same size?** The top-$k$ strategy enhances the performance by running a selected subset of the solver zoo for each instance, which certainly costs more time than individual solvers. For a fair comparison, we benchmark our top-$k$ selection method against a solver portfolio of the same size $k$. We construct this solver portfolio by exhaustively enumerating all possible subsets of size $k$ and selecting the one with the best overall performance. As shown in Appendix A.5, our top-$k$ selection consistently outperforms the size-$k$ solver portfolio across $k = \{1, 2, 3, 4\}$ on all datasets, i.e., TSP, CVRP, TSPLIB and CVRPLIB Set-X, demonstrating the effectiveness of our selection model.

**RQ2: How does the proposed selection framework perform when the problem distribution shifts and the problem scale increases?** We evaluate the generalization performance on two benchmarks, TSPLIB and CVRPLIB Set-X, which contain out-of-distribution and larger-scale instances. As shown in Table 2, all implementations of our selection framework generalize well, where the ranking model using top-$k$ selection improves the optimality gap by 0.88% (i.e., 1.95%-1.07%) on TSPLIB and by 0.71% (i.e., 6.10%-5.39%) on CVRPLIB Set-X, compared to the best individual solvers T2T and ELG on these two benchmarks. These results show that our selection framework is robust against the distribution shifts and increases in problem scale.

**RQ3: How do different implementations affect performance?** We evaluate and compare different implementations of the three components in our framework:

Table 2: Generalization results to TSPLIB and CVRPLIB Set-X datasets, which contain real-world out-of-distribution instances with larger scales.

| Methods | TSPLIB | | Methods | CVRPLIB Set-X | |
|---|---|---|---|---|---|
| | Gap | Time | | Gap | Time |
| BQ (*3rd*) | 3.04% | 1.44s | BQ (*3rd*) | 10.31% | 2.60s |
| DISFUCO (*2nd*) | 2.13% | 1.44s | Omni (*2nd*) | 6.21% | 0.38s |
| T2T (*1st*) | 1.95% | 1.74s | ELG (*1st*) | 6.10% | 1.31s |
| Oracle | 0.89% | 9.14s | Oracle | 5.10% | 6.81s |
| *Selection by classification* | | | *Selection by classification* | | |
| Greedy | 1.54% (0.05%) | 1.33s (0.02s) | Greedy | 5.96% (0.12%) | 1.06s (0.08s) |
| Top-$k$ ($k = 2$) | 1.22% (0.10%) | 2.47s (0.02s) | Top-$k$ ($k = 2$) | 5.44% (0.08%) | 2.40s (0.25s) |
| Rejection (20%) | 1.42% (0.11%) | 1.54s (0.03s) | Rejection (20%) | 5.83% (0.12%) | 1.31s (0.09s) |
| Top-$p$ ($p = 0.5$) | 1.49% (0.11%) | 1.37s (0.02s) | Top-$p$ ($p = 0.8$) | 5.79% (0.09%) | 1.42s (0.17s) |
| *Selection by ranking* | | | *Selection by ranking* | | |
| Greedy | 1.33% (0.06%) | 1.28s (0.03s) | Greedy | 5.76% (0.04%) | 1.31s (0.10s) |
| Top-$k$ ($k = 2$) | **1.07% (0.03%)** | 2.48s (0.02s) | Top-$k$ ($k = 2$) | **5.39% (0.06%)** | 2.56s (0.13s) |
| Rejection (20%) | 1.26% (0.03%) | 1.51s (0.04s) | Rejection (20%) | 5.63% (0.05%) | 1.60s (0.08s) |
| Top-$p$ ($p = 0.5$) | 1.28% (0.04%) | 1.46s (0.06s) | Top-$p$ ($p = 0.8$) | 5.61% (0.03%) | 1.72s (0.08s) |

Table 3: Mean (standard deviation) of optimality gaps of different feature extraction methods. All the models are trained using ranking loss, and employ greedy selection.

| Datasets | Best solver | Manual | Attention encoder | Hierarchical encdoer |
|---|---|---|---|---|
| TSP | 2.33% | 1.97% (0.01%) | 1.87% (0.02%) | **1.86% (0.01%)** |
| CVRP | 6.82% | 5.49% (0.08%) | **5.30% (0.01%)** | 5.31% (0.01%) |
| TSPLIB | 1.95% | 1.83% (0.03%) | 1.45% (0.11%) | **1.33% (0.06%)** |
| CVRPLIB | 6.10% | 6.35% (0.06%) | 5.87% (0.06%) | **5.76% (0.04%)** |

(1) **Feature extraction methods.** We compare the manual features (Smith-Miles et al., 2010) (see Appendix A.4), graph attention encoder Kool et al. (2019), and hierarchical graph encoder in Table 3. All methods are trained using ranking loss, and we report the optimality gap with greedy selection. As shown in Table 3, even the simplest manual features perform well, achieving better results than the best individual solver across three datasets — TSP, CVRP, and TSPLIB. This further validates the effectiveness of our selection framework. Comparing the third and fourth columns, we observe that the graph attention encoder consistently outperforms manual features on all datasets, verifying the superiority of learned features. Furthermore, by comparing the fourth and fifth columns, we find that while the graph attention encoder has already been effective on synthetic datasets, introducing the hierarchical encoder can further improve generalization performance on out-of-distribution datasets, TSPLIB and CVRPLIB Set-X, which is quite important in practice. This enhanced generalization capability may be attributed to the hierarchical encoder's ability to leverage the inherent hierarchical structures in COPs. More ablation studies of the hierarchical encoder are provided in Appendix A.6.

(2) **Loss functions to train the selection model.** We can clearly observe from Tables 1 and 2 that the model trained with ranking loss generally outperforms the one trained with classification loss, particularly when employing top-$p$ selection or under out-of-distribution settings. We also compare their accuracy of selecting the best single individual, which is similar as shown Appendix A.8. Thus, the benefit of ranking loss over classification loss shows the importance of incorporating the dominance relationships among sub-optimal solvers, which can make the selection more robust.

(3) **Selection strategies.** Greedy selection is efficient by selecting only the predicted best solver. Instead, top-$k$ selection selects the best $k$ solvers for better optimality gaps, but resulting in longer time. Rejection-based and top-$p$ selection provide a trade-off between optimality gap and time. Here, we focus on the evaluation of rejection-based and top-$p$ selection. We tune their hyperparameters (e.g., rejection ratio, $k$, and $p$) to obtain a range of results, provided in Appendix A.7. The results show the rejection-based selection with smaller $k$ ($k = 2$ or 3) tends to achieve better trade-off. Comparing top-$p$ selection and rejection-based selection, their performance has no significant difference. This is expected, because both of their principles are running more individual neural solvers when the confidence of the selection model is insufficient. However, the top-$p$ selection may be preferable in practice, where only one hyperparameter $p$ is associated.

## 5 CONCLUSION AND DISCUSSIONS

In this paper, we propose a general framework for neural solver selection for the first time, which can effectively select suitable solvers for each instance, leading to significantly better performance with little additional computational time, as validated by the extensive experiments on two well-studied COPs, TSP and CVRP. Besides TSP and CVRP, our proposed selection framework is adaptable to other problems. For new problems, one only needs to customize the feature extraction component. For instance, when adapting our framework to scheduling problems, one can adjust the graph attention encoder according to MatNet (Kwon et al., 2021) (i.e., add edge embeddings). We hope this preliminary work can open a new line for the topic of neural combinatorial optimization. Within the proposed selection framework, we preliminarily investigate several implementations of the three key components: Feature extraction, training loss functions, and selection strategies. Techniques such as hierarchical graph encoder, ranking loss, rejection-based selection, and top-$p$ selection notably enhance overall performance. Beyond the techniques presented, we discuss several promising avenues for further research under this framework.

**Feature extraction for neural solvers.** In our implementation, we only extracted features for problem instances and used fixed indices for neural solvers, which assumes a static neural solver zoo and can not directly utilize any newly added neural solver during deployment. To enable zero-shot generalization to unseen neural solvers, it is essential to construct a smooth feature space for solvers, where those with similar preferences and biases are positioned closely together. Here we design a preliminary method for extracting features of neural solvers to facilitate generalization to unseen solvers. This method is based on the insight that a neural solver's preferences can be characterized by representative instances where it significantly outperforms other solvers. For each neural solver, we sort those instances where it performs the best in an ascending order according to the ratio of the objective value that the solver obtains to the runner-up objective value, and select top 1% as its representative instances. Each representative instance is then treated as a token, and we apply a transformer to learn a summary feature from these instance tokens, which serves as the feature representation of the neural solver. Detailed implementations and results are provided in Appendix A.9. The results show that the preliminary method enables generalization to unseen neural solvers, where adding an extra solver can improve the selection performance.

Furthermore, advanced neural solver features should provide richer and deeper information than only using instance features to increase the capacity of the selection model. However, our preliminary method is based on the representative instances and fails to provide deeper information into the solvers' internal mechanisms. For future improvements, some approaches may be worth exploring, such as utilizing large language models to encode the code of neural solvers (Wu et al., 2024) or learning neural representations from their trained parameters (Kofinas et al., 2024), which can access internal solver information, and potentially improving selection performance.

**Runtime-aware selection for learn-to-seach solvers.** In this paper, since the average runtime of most individual neural solvers is short (approximately 1–2 seconds), we ignored their time difference during the training of the selection model, and only used the objective values obtained by the neural solvers as supervision information (by classification or ranking). However, if there are some time-consuming learn-to-search solvers, such as NeuOpt (Ma et al., 2021; 2023) and local reconstruction methods (Kim et al., 2021; Ye et al., 2024), in the solver pool, the runtime should be considered in the performance ranking. In such cases, developing a runtime-aware selection method to balance computational time and solution optimality would be necessary. To address this, we could penalize objective values based on time consumption or simultaneously optimize both metrics using multi-objective learning methods (Lin et al., 2022).

**Enhance the neural solver zoo by training.** As shown in Figure 6, current neural solvers can exhibit complementary performance over instances without any modification, which has motivated our framework of neural solver selection. Inspired by the population-based training (Grinsztajn et al., 2023), we can further enhance their complementary ability through finetuning, i.e., each neural solver is finetuned on those instances where it performs the best. We can also train new solvers from scratch by maximizing their performance contribution to the current solver zoo and iteratively add such new solvers for enhancement. Moreover, to facilitate the training and deployment of a neural solver zoo, it is essential to develop a unified platform that provides interfaces for executing and training diverse neural solvers, such as an extension to the existing RL4CO (Berto et al., 2024).

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

# A APPENDIX

## A.1 GRAPH ATTENTION LAYER

The graph attention layer is composed of two sub-layers: a multi-head attention sub-layer (Vaswani et al., 2017) and a feed-forward sub-layer. Each sub-layer is equipped with residual connection (He et al., 2016) and ReZero normalization (Bachlechner et al., 2021) for stable convergence of training. Denote the embedding of the $i$-th node as $\boldsymbol{h}_i$ (i.e., the $i$-th row of $H$). Since the graphs of TSP and CVRP are typically considered to be fully connected, the graph attention layer is calculated as

$$\hat{\boldsymbol{h}}_i = \boldsymbol{h}_i^{l-1} + \alpha^l \mathrm{MHA}_i^l(\boldsymbol{h}_1^{l-1}, \boldsymbol{h}_2^{l-1}, ..., \boldsymbol{h}_N^{l-1}),$$
$$\boldsymbol{h}_i^l = \hat{\boldsymbol{h}}_i + \alpha^l \mathrm{FF}(\hat{\boldsymbol{h}}_i),$$

where $l$ is the layer index, $i$ is the node index, $\alpha^l$ is a learnable parameter used in the ReZero normalization, MHA and FF are short for the multi-head attention and the feed-forward network, respectively. For the implementations of the basic components MHA and FF, we refer to Vaswani et al. (2017) for details. Specifically, following the common settings of previous works, we set the dimension of $\boldsymbol{h}$ to 128, the number of heads in MHA to 8, and the hidden dimension of FF to 512.

## A.2 DATA GENERATION

Synthetic TSP and CVRP instances are generated for training, where the node coordinates, demands, and vehicle capacities are all sampled from manually defined distributions. The scale of each instance is sampled from $[50, 500]$ randomly. Details of node coordinates, vehicle capacity, and node demands are introduced as follows.

**Node coordinates** To generate diverse training instances, we utilize Gaussian mixture distributions to sample the node coordinates for both TSP and CVRP, which is common in previous works (Manchanda et al., 2022; Zhou et al., 2023) and demonstrates effectiveness on approximating various node distributions with different hardness levels (Smith-Miles et al., 2010). First, we randomly select the number of Gaussian components $c \sim U(0, 15)$ (when $c = 0$, we use the uniform distribution) and partition the nodes randomly into $c$ groups, one for each component. For each Gaussian component, we sample the mean coordinates $\boldsymbol{\mu} = (x_\mu, y_\mu)$ by $x_\mu \sim U(0, 1)$ and $y_\mu \sim U(0, 1)$, and sample the variances $\mathrm{var}_x$ and $\mathrm{var}_y$ uniformly from $[1, 100]$. The covariance cov is sampled uniformly from $[-\sqrt{\mathrm{var}_x \cdot \mathrm{var}_y}, \sqrt{\mathrm{var}_x \cdot \mathrm{var}_y})$, forming the covariance matrix $\Sigma = \begin{bmatrix} \mathrm{var}_x & \mathrm{cov} \\ \mathrm{cov} & \mathrm{var}_y \end{bmatrix}$. Node coordinates are sampled from the $N(\boldsymbol{\mu}, \Sigma)$ and then scaled to the square of $x, y \in [0, 1]$. Unlike conventional Gaussian mixture distributions (Manchanda et al., 2022; Zhou et al., 2023), which often use an identity covariance matrix, our approach employs randomized covariance matrices $\Sigma$. This modification can produce more diverse instances by introducing more variability in the node distributions.

**Vehicle capacity and node demands** We employ two vehicle capacity distributions to generate CVRP instances: (1) Scale-related distribution (Zhou et al., 2023): The vehicle capacity is proportional to the scale $N$, defined as $Q = 30 + \lceil \frac{N}{5} \rceil$. (2) Triangular distributions (Uchoa et al., 2017): The parameters of the triangular distribution include the upper limit $ub$, mode $m$, and lower limit $lb$, which are randomly sampled in succession as follows: $ub \sim U(20, \frac{N}{2})$, $m \sim U(5, ub)$, and $lb \sim U(3, m)$. The triangular distribution $T(lb, m, ub)$ is then used to generate vehicle capacities, resulting in more diverse CVRP instances compared to the fixed capacity setting (Nazari et al., 2018). Each capacity distribution is selected with equal probability. Node demands $m_i$ are sampled uniformly from $U(1, 10)$ and normalized by dividing by $Q$.

## A.3 ELIMINATE USELESS NEURAL SOLVERS

The preserved neural solvers should have distinct strengths in certain problem instances, ensuring that they can bring significant improvements in overall performance. Motivated by this, we propose a simple yet effective heuristic strategy to build the neural solver zoo based on the assessment of their contribution to the overall performance.

Given the alternative neural solvers $\boldsymbol{S} = \{s_1, s_2, s_3, ....\}$, we assess the contribution of a specific solver $s_i \in \boldsymbol{S}$ by the degradation of performance after removing it. That is, the assessment of $s_i$ can be formalized as

$$\mathcal{A}(s_i) = \mathbb{E}_I \left[ \mathcal{P}_I(\boldsymbol{S}) - \mathcal{P}_I(\boldsymbol{S}/s_i) \right],$$

where $\mathcal{P}_I(\cdot)$ denotes the performance of a neural solver zoo on instance $I$. Here we use the percentage of the optimality gap to define $\mathcal{P}_I(\cdot)$ and employ a validation set for the estimation of expectations. According to this criteria, we can estimate the alternative solvers and remove the one with the lowest assessed contribution from $\boldsymbol{S}$. This process repeats iteratively until for all $s_i \in \mathcal{S}$, $\mathcal{A}(s_i)$ surpasses the predefined threshold $\delta$, indicating the significance of each alternative neural solver. In practice, we collect the prevailing competitive neural solvers in the community to compose the original $\mathcal{S}$ and set $\delta$ as $0.01\%$.

The neural solver zoos before and after elimination are listed in Table 4. Note that for DIFUSCO and T2T, multiple models are released. We collect both the models trained on the N = 100 dataset and the N = 500 dataset as alternatives simultaneously.

Table 4: The neural solver zoo before and after elimination.

| Stage | Neural solver zoo for TSP | Neural solver zoo for CVRP |
|---|---|---|
| Before elimination | BQ, LEHD, Omni, ELG, INViT, DIFUSCO (N=100), DIFUSCO (N=500), T2T (N=100), T2T (N=500) | BQ, LEHD, Omni, ELG, INViT, MVMoE |
| After elimination | BQ, LEHD, ELG, DIFUSCO (N=100), DIFUSCO (N=500), T2T (N=100), T2T (N=500) | BQ, LEHD, Omni, ELG, MVMoE |

## A.4 MANUAL FEATURES

We reproduce the manual features proposed by Smith-Miles et al. (2010), which use statistical information and cluster analysis results to describe the characteristics of TSP. In this paper, we adopt these features: the standard deviation of the distances, the coordinates of the instance centroid, the radius of the TSP instance, the fraction of distinct distances, the variance of the normalized nearest neighbour distances (nNNd's), the coefficient of variation of the nNNd's, the ratio of the number of clusters to the number of nodes (Here we use HDBSCAN algorithm (Campello et al., 2013) to generate clusters), the ratio of number of outliers to nodes, and the mean radius of the clusters. For CVRP, we further add the mean and standard deviation of node demands to the features.

## A.5 RESULTS OF TOP-$k$ SELECTION.

We compare the performance of our top-$k$ selection and the solver portfolio with the same size $k$ on four datasets, including TSP, CVRP, TSPLIB and CVRPLIB Set-X. As shown in Figure 3, our top-$k$ selection consistently outperforms the size-$k$ solver portfolio across $k \in \{1, 2, 3, 4\}$. We also observe that the performance of our top-$k$ selection is close to the Oracle when $k = 4$. Moreover, it is expected that the performance improvement of our top-$k$ selection gradually diminishes as $k$ increases, since the performance of solver portfolio is also approaching the Oracle (the gray line).

Related works, such as ZTop (Bai et al., 2021), employ a fixed set of neural solvers to construct a portfolio for all instances, resembling the static portfolio approach compared in this study. In contrast, our top-$k$ selection strategy dynamically constructs instance-specific portfolios, offering greater flexibility and a higher potential for performance improvement. As demonstrated in Figure 3, our method consistently outperforms the static portfolio approach across all portfolio sizes.

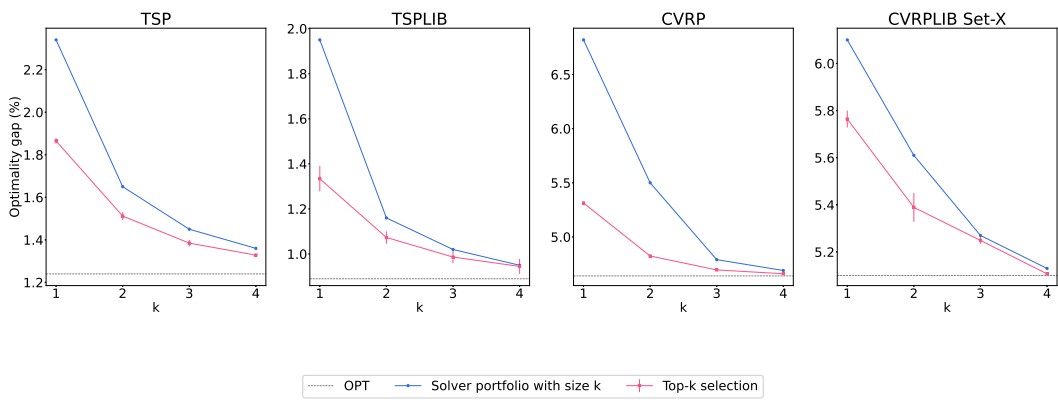

Figure 3: Comparisons of the proposed top-$k$ selection and the solver portfolio with size $k$.

A.6 ABLATION OF THE HIERARCHICAL GRAPH ENCODER.

The proposed hierarchical graph encoder utilizes graph pooling to downsample the instance graph and aggregates features obtained from multiple levels of the downsampled graphs. To evaluate the effectiveness of the graph pooling, we employ a graph encoder that aggregates the features from multiple layers for comparison, which is a clear ablation study since the main difference is that it does not have the graph pooling operation.

The results in Table 5 show that using our hierarchical graph encoder outperforms the encoder that simply accumulates multi-layer features, especially in terms of the generalization performance on CVRPLIB Set-X. This demonstrates the effectiveness of the graph pooling operation.

Table 5: Ablation study of hierarchical graph encoder. We report the mean and standard deviation of five independent runs. All the models are trained using ranking loss, and employ greedy selection.

| Datasets | Attention encoder | + Multi-layer features | Hierarchical encoder |
|---|---|---|---|
| TSP | 1.87% (0.02%) | 1.87% (0.01%) | **1.86% (0.01%)** |
| CVRP | **5.30% (0.01%)** | 5.30% (0.02%) | 5.31% (0.01%) |
| TSPLIB | 1.45% (0.11%) | 1.35% (0.05%) | **1.33% (0.06%)** |
| CVRPLIB | 5.87% (0.06%) | 5.86% (0.08%) | **5.76% (0.04%)** |

To evaluate the computational efficiency of the hierarchical encoder, we provide detailed comparisons of the computation cost and optimality between our hierarchical encoder and a typical graph encoder. The results are shown in Table 6, which includes the inference time per instance on TSPLIB, training time per epoch, and the average optimality gap on TSPLIB.

Table 6: Comparisons of the computation cost and optimality between our hierarchical encoder and a typical attention encoder.

| Methods | Inference time of selection model | Inference time of neural solvers | Training time each epoch | Optimality gap |
|---|---|---|---|---|
| Naive attention encoder | 0.0054s | 1.2600s | 1m40s | 1.54% |
| Hierarchical encoder | 0.0070s | 1.2961s | 2m30s | 1.37% |

We can observe from the second column that the introduction of our hierarchical encoder will increase the inference time of the selection model a little bit, e.g., from 0.0054s to 0.0070s. However, as shown in the second and third columns, the inference time of the selection model is orders of magnitude shorter than that of the neural solvers, so the inference efficiency of the selection model is less of a concern. The fourth column shows that the training time per epoch of the naïve encoder and the hierarchical encoder are 1m40s and 2m30s, respectively. Although the hierarchical encoder slows the training, the total runtime for 50 epochs is still only 2 hours, which is acceptable in most scenarios. Therefore, the performance metric (i.e., optimality gap) of different encoders is more crucial, especially the generalization performance. If the encoder learns robust representations, we can directly transfer the selection model to different datasets in a zero-shot manner, saving the time for fine-tuning and adaptation. Considering the better generalization (e.g., the optimality gap decreases from 1.54% to 1.37%), we believe that the proposed hierarchical encoder is a better choice.

A.7 DETAILED COMPARISONS OF SELECTION STRATEGIES

According to the mechanisms of the four selection strategies, they have different preferences in the trade-off of efficiency and optimality. Generally, for efficiency, Greedy > Rejection ≈ Top-$p$ > Top-$k$, for optimality, Top-$k$ > Rejection ≈ Top-$p$ > Greedy. Meanwhile, the hyper-parameters of them can be used for balancing efficiency and optimality as well. As a result, the choice of different selection strategies can be decided by the users according to their preference, and we suggest using Top-p or Rejection as the default choices since they can adaptively select solvers based on the confidence of the selection model.

The rejection-based selection and top-$p$ selection are both designed to achieve better performance with little additional time consumption. To evaluate them in detail, we tune their parameters (e.g.,

rejection ratio, $k$, and $p$) to obtain a range of results. For the rejection-based selection, we use $k \in \{2, 3, 4\}$ and vary the rejection ratio from 0.05 to 0.85 in increments of 0.05. For the top-$p$ selection, we adjust the value of $p$ from 0.40 to 0.95 in increments of 0.01. The results of the optimality gap and time consumption are provided in Figure 4. As shown in the figures, the rejection strategy with smaller $k$ tends to achieve better optimality gaps using the same time consumption. Therefore, we recommend $k = 2$ or $3$ when using rejection-based selection. Comparing top-$p$ selection and rejection-based selection, we can not definitively conclude which strategy is superior, which is expected since they share a similar idea of utilizing confidence levels to decide whether to employ multiple solvers. However, the top-$p$ selection may be preferable in practice due to its simplicity, where only a single hyperparameter $p$ requires tuning.

Furthermore, these results highlight $p$ and the rejection ratio as important hyperparameters that allow users to balance efficiency and optimality. Though using a fixed value has led to good performance in our experiments, we believe that adaptively adjusting $p$ and rejection ratio for each instance could further improve performance. Therefore, it is interesting to develop an instance-specific adapter for $p$ or rejection ratio by leveraging instance features, which we leave for future works.

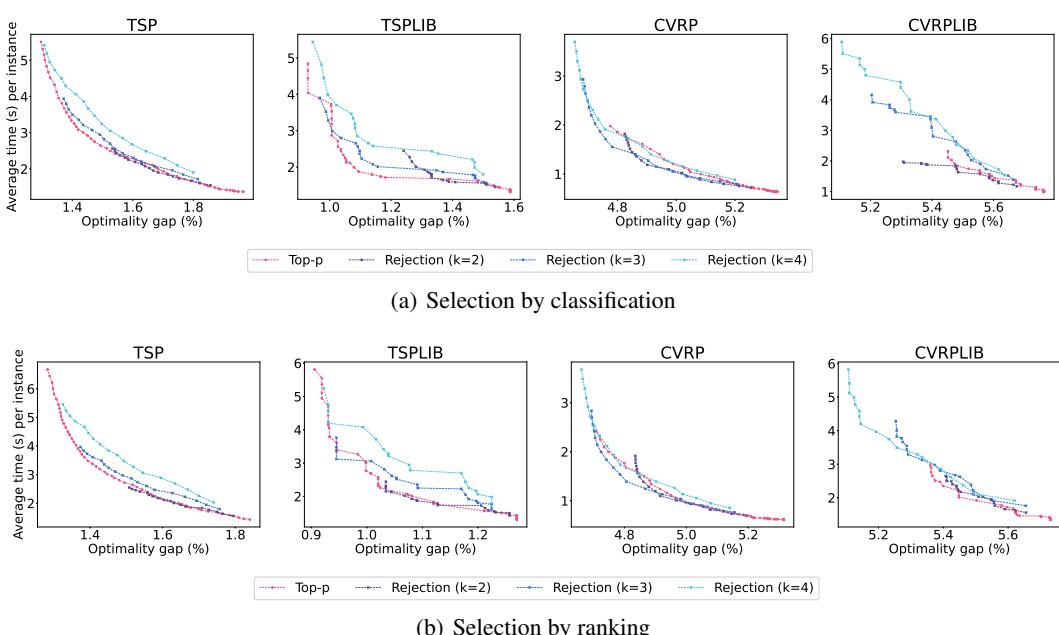

Figure 4: Performance of the rejection-based and top-$p$ selection.

## A.8 Selection accuracy

We present the accuracy of selecting the optimal neural solver in Table 7. The results show that the ranking model and classification model generally have similar selection accuracy, except that the ranking model achieves better accuracy than the classification model on CVRPLIB.

Table 7: Accuracy of models trained by different losses using greedy selection. We report the mean and standard deviation of five independent runs.

| Metrics | Classification | Ranking |
|---|---|---|
| Accuracy on TSP | 36% (1%) | 35% (1%) |
| Accuracy on CVRP | 61% (1%) | 62% (0%) |
| Accuracy on TSPLIB | 40% (3%) | 40% (7%) |
| Accuracy on CVRPLIB | 52% (2%) | 56% (3%) |

A.9 NEURAL SOLVER FEATURES

To enable generalization to unseen neural solvers, we design a preliminary feature extraction method for neural solvers, which utilizes representative instances to represent a neural solver. For each neural solver, we sort those instances where the neural solver performs the best in an ascending order according to the ratio of the objective value that the solver obtains to the runner-up objective value, and select the top 1% as its representative instances. Then, we use an instance encoder to obtain embeddings for each representative instance, serving as their token vectors. A transformer with two layers is employed to learn a summary feature from these instance tokens, which serves as the feature representation of the neural solver. More details are described as follows.

**Instance tokenization.** We use a hierarchical graph encoder as the tokenization encoder to generate embeddings for each representative instance. Note that the parameters $\theta'$ of the tokenization encoder are not updated by back-propagation. Instead, to stabilize the instance tokens during training, we update $\theta'$ using a momentum-based moving average of the parameters $\theta$ of the instance feature encoder: $\theta' \leftarrow m \cdot \theta' + (1 - m) \cdot \theta$, where $m \in [0, 1)$ is a momentum coefficient (We set $m = 0.99$ in experiments). Only the parameters $\theta$ are updated via back-propagation. This momentum update ensures that $\theta'$ evolves more smoothly than $\theta$, resulting in stable instance tokenization.

**Transformer architecture.** For each neural solver, we utilize the tokens of its representative instances along with a learnable summary token to compute a summary representation. We apply two attention layers for this purpose. The first layer is a self-attention mechanism applied over all tokens (including the summary token), enabling interactions among them. The second attention layer uses only the summary token as the query and all tokens as keys and values, effectively aggregating information from all tokens into the summary token. The final embedding of the summary token is then output as the neural solver's representation.

**Selection model with neural solver feature.** The selection model integrates both the instance features and the neural solver features to output a score for each instance-solver pair. We employ an MLP to compute these scores. For each instance, the scores across all neural solvers are normalized to derive the probability distribution for solver selection.

To integrate a newly added neural solver, we first identify its representative instances from the training dataset and employ the aforementioned networks to compute its feature representation. The selection model can then leverage this new solver by considering its feature during selection, without the need for any fine-tuning.

To evaluate the effectiveness of this method, we remove the second-best neural solver from the current solver zoo, train the selection model using the supervision information provided by the pruned solver zoo, and reintroduce the removed solver during testing. Figure 5 presents the top-$k$ selection performance with and without the newly added (extra) solver. The results show that the performance with the newly added solver is generally better than the performance without it, demonstrating that the selection model can leverage the information of unseen solvers without any finetuning. In other words, the selection model can generalize to unseen solvers. However, we observe a slight decrease in top-1 performance, indicating that the preliminary method requires further improvement.

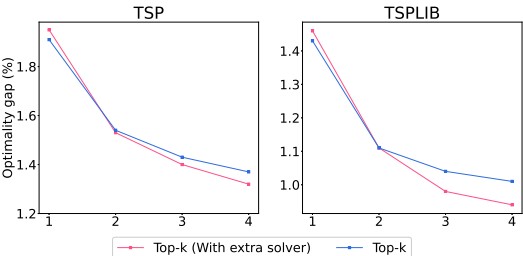

Figure 5: Performance of introducing an extra neural solver based on the neural solver feature. The results of top-$k$ selection with and without extra neural solver are presented.

## A.10 COMPLETE RESULTS WITH ALL NEURAL SOLVERS

Table 8: Empirical results on synthetic TSP and TSPLIB datasets, reporting the mean (standard deviation) over five independent runs. **All individual solvers** are included for comparison, and Oracle denotes the optimal performance for selection, which is computed by running all individual solvers in the zoo for each instance and selecting the best one. The best individual solver and its results are underlined, and the best optimality gaps, excluding Oracle, are highlighted in boldface. The suffixes '-N100' and '-N500' indicate models trained on datasets with N=100 and N=500, respectively. Obj denotes the average objective value on the dataset.

| Methods | TSP | | |
| --- | --- | --- | --- |
| | Obj | Gap | Time |
| BQ | 8.13 | 3.00% | 1.61s |
| ELG | 8.18 | 3.70% | 0.45s |
| LEHD | 8.17 | 3.57% | 0.86s |
| T2T-N100 | 8.08 | 2.48% | 1.71s |
| T2T-N500 | 8.08 | 2.40% | 1.98s |
| DIFUSCO-N100 | 8.11 | 2.84% | 1.46s |
| DIFUSCO-N500 | 8.07 | 2.33% | 1.45s |
| Oracle | 7.99 | 1.24% | 8.93s |
| *Selection by classification* | | | |
| Greedy | 8.04 (0.00) | 1.94% (0.02%) | 1.36s (0.01s) |
| Top-$k$ ($k = 2$) | 8.01 (0.00) | 1.53% (0.01%) | 2.52s (0.04s) |
| Rejection (20%) | 8.03 (0.00) | 1.81% (0.01%) | 1.63s (0.01s) |
| Top-$p$ ($p = 0.5$) | 8.03 (0.00) | 1.84% (0.03%) | 1.55s (0.06s) |
| *Selection by ranking* | | | |
| Greedy | 8.04 (0.00) | 1.86% (0.01%) | 1.33s (0.01s) |
| Top-$k$ ($k = 2$) | **8.01 (0.00)** | **1.51% (0.02%)** | 2.56s (0.03s) |
| Rejection (20%) | 8.03 (0.00) | 1.75% (0.02%) | 1.63s (0.01s) |
| Top-$p$ ($p = 0.5$) | 8.02 (0.00) | 1.68% (0.02%) | 1.86s (0.07s) |

| Methods | TSPLIB | | |
| --- | --- | --- | --- |
| | Obj | Gap | Time |
| BQ | 8.29 | 3.04% | 1.44s |
| ELG | 8.29 | 3.05% | 0.40s |
| LEHD | 8.26 | 2.57% | 0.88s |
| T2T-N100 | 8.22 | 2.09% | 1.76s |
| T2T-N500 | 8.21 | 1.95% | 1.74s |
| DISFUCO-N100 | 8.23 | 2.25% | 1.44s |
| DISFUCO-N500 | 8.22 | 2.13% | 1.44s |
| Oracle | 8.12 | 0.89% | 9.14s |
| *Selection by classification* | | | |
| Greedy | 8.17 (0.00) | 1.54% (0.05%) | 1.33s (0.02s) |
| Top-$k$ ($k = 2$) | 8.15 (0.01) | 1.22% (0.10%) | 2.47s (0.02s) |
| Rejection (20%) | 8.16 (0.01) | 1.42% (0.11%) | 1.54s (0.03s) |
| Top-$p$ ($p = 0.5$) | 8.17 (0.01) | 1.49% (0.11%) | 1.37s (0.02s) |
| *Selection by ranking* | | | |
| Greedy | 8.16 (0.00) | 1.33% (0.06%) | 1.28s (0.03s) |
| Top-$k$ ($k = 2$) | 8.14 (0.00) | **1.07% (0.03%)** | 2.48s (0.02s) |
| Rejection (20%) | 8.15 (0.00) | 1.26% (0.03%) | 1.51s (0.04s) |
| Top-$p$ ($p = 0.5$) | 8.15 (0.00) | 1.28% (0.04%) | 1.46s (0.06s) |

Table 9: Empirical results on synthetic CVRP and CVRPLIB Set-X datasets, reporting the mean (standard deviation) over five independent runs. **All individual solvers** are included for comparison. Obj denotes the average objective value on the dataset.

| Methods | CVRP | | |
| --- | --- | --- | --- |
| | Obj | Gap | Time |
| BQ | 18.39 | 7.20% | 1.59s |
| ELG | 18.49 | 7.81% | 0.82s |
| LEHD | 18.42 | 7.37% | 1.01s |
| MVMoE | 19.48 | 13.56% | 0.70s |
| Omni | 18.32 | 6.82% | 0.24s |
| Oracle | 17.95 | 4.64% | 4.38s |
| *Selection by classification* | | | |
| Greedy | 18.07 (0.00) | 5.35% (0.02%) | 0.64s (0.01s) |
| Top-$k$ ($k = 2$) | **17.98 (0.00)** | **4.81% (0.01%)** | 1.87s (0.03s) |
| Rejection (20%) | 18.04 (0.01) | 5.19% (0.03%) | 0.77s (0.01s) |
| Top-$p$ ($p = 0.8$) | 18.04 (0.01) | 5.16% (0.03%) | 0.87s (0.08s) |
| *Selection by ranking* | | | |
| Greedy | 18.06 (0.00) | 5.31% (0.01%) | 0.62s (0.01s) |
| Top-$k$ ($k = 2$) | 17.98 (0.00) | 4.82% (0.01%) | 1.90s (0.04s) |
| Rejection (20%) | 18.04 (0.00) | 5.15% (0.02%) | 0.74s (0.01s) |
| Top-$p$ ($p = 0.8$) | 18.01 (0.00) | 4.99% (0.02%) | 1.03s (0.03s) |

| Methods | CVRPLIB Set-X | | |
| --- | --- | --- | --- |
| | Obj | Gap | Time |
| BQ | 71.21 | 10.31% | 2.60s |
| ELG | 68.50 | 6.10% | 1.31s |
| LEHD | 73.40 | 13.70% | 1.60s |
| MVMoE | 74.59 | 15.54% | 0.90s |
| Omni | 68.57 | 6.21% | 0.38s |
| Oracle | 67.85 | 5.10% | 6.81s |
| *Selection by classification* | | | |
| Greedy | 68.41 (0.08) | 5.96% (0.12%) | 1.06s (0.08s) |
| Top-$k$ ($k = 2$) | 68.07 (0.05) | 5.44% (0.08%) | 2.40s (0.25s) |
| Rejection (20%) | 68.32 (0.08) | 5.83% (0.12%) | 1.31s (0.09s) |
| Top-$p$ ($p = 0.8$) | 68.30 (0.06) | 5.79% (0.09%) | 1.42s (0.17s) |
| *Selection by ranking* | | | |
| Greedy | 68.28 (0.03) | 5.76% (0.04%) | 1.31s (0.10s) |
| Top-$k$ ($k = 2$) | **68.04 (0.04)** | **5.39% (0.06%)** | 2.56s (0.13s) |
| Rejection (20%) | 68.19 (0.03) | 5.63% (0.05%) | 1.60s (0.08s) |
| Top-$p$ ($p = 0.8$) | 68.18 (0.02) | 5.61% (0.03%) | 1.72s (0.08s) |

## A.11 SEPARATE RESULTS OF DIFFERENT SCALES

We provide separate results of different scales for a deeper investigation, where we divide the $N \in [50, 500]$ to four subsets as shown in the first row of Table 10. The results in Tables 10 and 11 demonstrate that our selection method consistently outperforms the single best solver across different problem scales on both TSP and CVRP datasets.

Table 10: Separate results according to problem scale $N$ on synthetic TSP dataset. We report the mean (standard deviation) optimality gap over five independent runs.

| Methods / $N$ | $[50, 200]$ | $(200, 300]$ | $(300, 400]$ | $(400, 500]$ |
|---|---|---|---|---|
| Single best solver | 0.96% | 2.34% | 2.78% | 2.98% |
| Oracle | 0.39% | 1.19% | 1.70% | 2.18% |
| Ours (Greedy) | 0.84% (0.03%) | 2.01% (0.02%) | 2.43% (0.02%) | 2.71% (0.03%) |
| Ours (Top-$k$, $k = 2$) | 0.61% (0.02%) | 1.53% (0.03%) | 1.99% (0.03%) | 2.41% (0.05%) |
| Ours (Rejection, 20%) | 0.75% (0.04%) | 1.86% (0.04%) | 2.33% (0.03%) | 2.62% (0.02%) |
| Ours (Top-$p$, $p = 0.5$) | 0.71% (0.02%) | 1.70% (0.02%) | 2.24% (0.04%) | 2.57% (0.04%) |

Table 11: Separate results according to problem scale $N$ on synthetic CVRP dataset. We report the mean (standard deviation) optimality gap over five independent runs.

| Methods / $N$ | $[50, 200]$ | $(200, 300]$ | $(300, 400]$ | $(400, 500]$ |
|---|---|---|---|---|
| Single best solver | 3.95% | 6.06% | 7.76% | 9.24% |
| Oracle | 2.17% | 4.33% | 5.74% | 7.40% |
| Ours (Greedy) | 2.85% (0.03%) | 4.87% (0.02%) | 6.47% (0.05%) | 8.09% (0.01%) |
| Ours (Top-$k$, $k = 2$) | 2.32% (0.02%) | 4.54% (0.02%) | 5.91% (0.03%) | 7.55% (0.03%) |
| Ours (Rejection, 20%) | 2.64% (0.02%) | 4.70% (0.03%) | 6.22% (0.02%) | 7.91% (0.03%) |
| Ours (Top-$p$, $p = 0.5$) | 2.36% (0.02%) | 4.70% (0.04%) | 6.21% (0.05%) | 7.81% (0.02%) |

## A.12 ADDITIONAL RESULTS FOR MORE NEURAL SOLVERS AND LARGER-SCALE DATASETS

We add several new solvers to our pool, increase the problem scale from $N \in [50, 500]$ to $N \in [500, 2000]$, and use the enhanced solver pool to conduct new experiments. The results shown in Table 12 demonstrate that our framework can be compatible with more neural solvers and can also improve performance over the single best solver on larger-scale instances.

For details, we add two divide-and-conquer solvers, GLOP Ye et al. (2024) and UDC Zheng et al. (2024), to our solver pool, which can significantly enhance the overall performance. The construction of our solver pool now considers reinforced (ELG, INViT), supervised (BQ, LEHD), meta-learning-based (Omni), diffusion-based (DIFUSCO, T2T), and divide-and-conquer (GLOP, UDC) methods. The experimental results have shown that our proposed framework can effectively combine the advantages of these neural solvers and significantly improve performance.

Table 12: Experimental results on the larger-scale instances with $N \in [500, 2000]$. We report the mean (standard deviation) over five independent runs.

| Methods | Synthetic TSP with $N \in [500, 2000]$ | |
|---|---|---|
| | Gap | Time |
| Single best solver | 6.104% | 8.369s |
| Ours (Greedy) | 5.540% (0.038%) | 8.322s (0.036s) |
| Ours (Top-$k$, $k = 2$) | 5.369% (0.003%) | 15.566s (0.085s) |
| Single best of new solver pool | 3.562% | 5.274s |
| Ours with new solvers (Greedy) | 3.126% (0.002%) | 6.892s (0.006s) |
| Ours with new solvers (Top-$k$, $k = 2$) | 2.955% (0.005%) | 13.713s (0.036s) |

## A.13 COMPARISONS WITH TRADITIONAL ALGORITHM SELECTION METHODS

To further demonstrate the effectiveness of our proposed techniques, we provide additional comparison results between our proposed method and existing algorithm selection methods for non-neural TSP solvers (Smith-Miles et al., 2010; Seiler et al., 2020), as shown in Table 13. In fact, the method of using features from Smith-Miles et al. (2010) and our ranking model was also compared in Table 3. The R package *salesperson*[2] provides the up-to-now most comprehensive collection of features for TSP and is widely used in algorithm selection methods (Seiler et al., 2020; Heins et al., 2021). Based on the feature set of *salesperson*, we reproduce an advanced algorithm selection method (Seiler et al., 2020) following the pipeline that computes hand-crafted features, conducts feature selection, and applies random forest for classification, where we employ the univariate statistical test to select important features. Besides, we also combine the *salesperson* features with our ranking model for ablation, denoted by "Seiler et al. (2020) + Ranking" in Table 13.

Table 13: Comparison experiments with algorithm selection methods for TSP. We report the mean (standard deviation) over five independent runs.

| Methods | Synthetic TSP | | TSPLIB | |
|---|---|---|---|---|
| | Gap | Time | Gap | Time |
| Single best solver | 2.33% | 1.45s | 1.95% | 1.74s |
| Oracle | 1.24% | 8.93s | 0.89% | 9.14s |
| *Algorithm selection methods* | | | | |
| Smith-Miles et al. (2010) + Ranking | 1.97% (0.01%) | 1.37s (0.01s) | 1.83% (0.03%) | 1.32s (0.05s) |
| Seiler et al. (2020) | 2.12% (0.04%) | 1.35s (0.00s) | 1.56% (0.01%) | 1.34s (0.05s) |
| Seiler et al. (2020) + Ranking | 1.95% (0.01%) | 1.33s (0.03s) | 1.55% (0.03%) | 1.27s (0.06s) |
| *Our selection method using ranking* | | | | |
| Greedy | 1.86% (0.01%) | 1.33s (0.01s) | 1.33% (0.06%) | 1.28s (0.03s) |
| Top-$k$ ($k = 2$) | **1.51% (0.02%)** | 2.56s (0.03s) | **1.07% (0.03%)** | 2.48s (0.02s) |
| Rejection (20%) | 1.75% (0.02%) | 1.63s (0.01s) | 1.26% (0.03%) | 1.51s (0.04s) |
| Top-$p$ ($p = 0.5$) | 1.68% (0.02%) | 1.86s (0.07s) | 1.28% (0.04%) | 1.46s (0.06s) |

The experimental results in Table 13 indicate that our proposed method can achieve superior performance than advanced algorithm selection methods on both synthetic TSP and TSPLIB. Comparing the fifth and sixth rows, our proposed hierarchical encoder demonstrates superior performance over the *salesperson* features, especially on the out-of-distribution benchmark TSPLIB. Additionally, the comparison of the fourth and fifth rows shows that our deep learning-based ranking model achieves better results than traditional classification methods. Furthermore, the results of the last three rows illustrate that our proposed adaptive selection strategies effectively enhance optimality with minimal increases in time consumption.

## A.14 EXPLANATION FOR THE DATASET CHOICE

Most NCO methods use specific benchmarks with fixed distributions and scales, like uniform TSP datasets, to evaluate the optimization and generalization ability under controlled conditions. Many prevailing methods have achieved excellent performance on these benchmarks (e.g., gap $< 0.5\%$ on uniform TSP100). In contrast, our study focuses on a harder setting by using a dataset with diverse instances of varying distributions and scales, where the properties of instances are not specified in advance. This approach allows us to assess whether a selection method can effectively identify the suitable solver for a wide range of instances.

Additionally, we also provide results of coordinating multiple neural solvers on the widely-used uniform TSP100, as shown in Table 14. These results show that while selection on this dataset can still be effective, the potential improvement over the best single solver is limited, as single solvers already perform well on the uniform dataset.

---

[2] https://github.com/jakobbossek/salesperson

Table 14: Results of coordinating multiple neural solvers on different datasets.

| Methods | Our synthetic TSP dataset | Unifrom TSP100 |
|---|---|---|
| Single best solver | 2.33% | 0.29% |
| Oracle | 1.24% | 0.10% |

## A.15 ILLUSTRATION OF THE NODES SAMPLED BY HIERARCHICAL GRAPH ENCODER

We illustrate the retained nodes after downsampling. Surprisingly, we can find some consistent patterns which are intuitively reasonable. We summarize them as three main points:

- Cluster nodes. As illustrated in Figures 6(a) and 6(b), when instances contain certain clusters, the hierarchical encoder tends to select a subset of "representative" nodes from each cluster, efficiently describing the entire spatial distribution.

- Specific blocks. As illustrated in Figures 6(c) and 6(d), when instances contain specific complex geometric patterns like squares (Figure 6(c)) and arrays (Figure 6(d)), the hierarchical encoder can capture the nodes of these important areas to identify their characteristics.

- Boundary nodes. For instances without clear sub-components, the hierarchical encoder tends to focus on boundary nodes that describe the global shape, as illustrated in Figures 6(e) and 6(f).

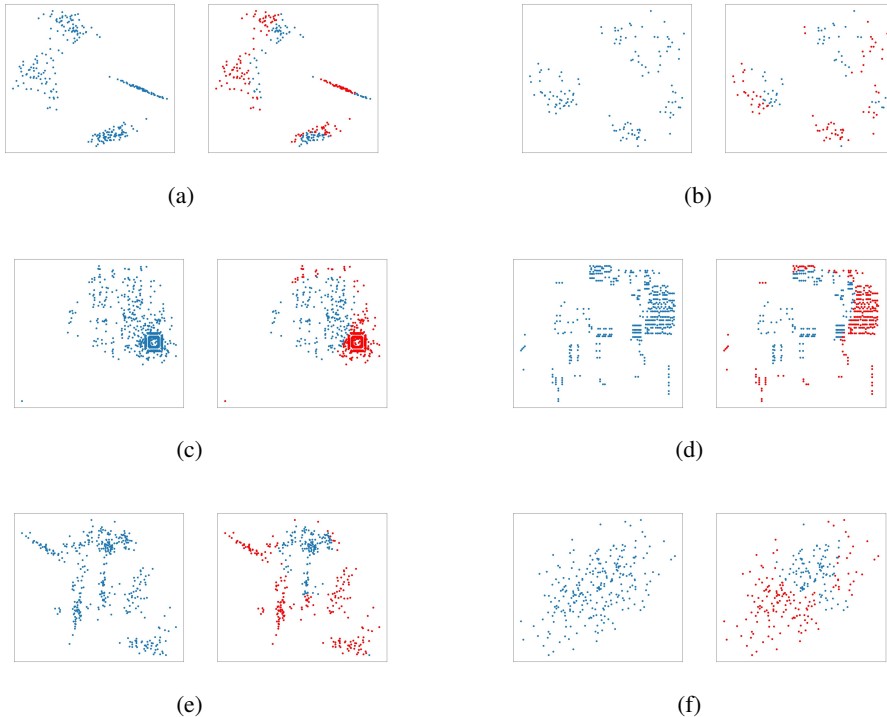

Figure 6: Illustrations of nodes selected by the hierarchical encoder. Each sub-figure represents an instance of TSP. The blue nodes represent the original instance, and the red nodes represent the retained nodes after down-sampling by the hierarchical encoder.

