# OpenReview forum: "Neural Solver Selection for Combinatorial Optimization"
_ICLR.cc/2025/Conference — Submitted to ICLR 2025_

### Official Review · Reviewer_SaPT · 2024-10-24

**Soundness:** 3
**Presentation:** 3
**Contribution:** 2
**Rating:** 5
**Confidence:** 5

**Summary:**

This paper introduces a framework for selecting the most appropriate neural solvers for TSP and CVRP at instances level. The framework enhances performance by allocating each problem instance to the most suitable solvers from a pool of available neural solvers via graph encoding and tailored selection model and strategy.

**Strengths:**

The paper is generally well-structured and easy to follow. The writing is clear and the presentation of the idea is concise. The idea of selecting neural solvers at instance level is interesting and of practical significance.

**Weaknesses:**

1. The novelty of this work is somewhat limited because the idea of ranking different existing solvers for individual instances is not technically innovative, and the framework seems to highly rely on previous solvers, graph encoders, as well as established losses and selection strategies.
2. To gain the supervision for training requires executing multiple solvers on the same training set, which is probably time-consuming. Furthermore, given such computational overhead, it is believed that a tediously sequential performing of them on the targeted dataset can have been already done for simple selection of the optimal result. Thus, further clarification is needed on the necessity of this proposal.
3. The OPT in the evaluation is somewhat misleading. I suggest the authors solving the test instances with exact solvers or powerful heuristics like Gurobi, LKH3, HGS, etc, as reference solution for the computation of optimality gaps, which also better aligns with previous works.
4. Additionally, adding such heuristics (in point 3) in your selection zoo is worth considering for further experimental results. If the neural solvers achieve comparable performance as the learning-free methods, the significance of this work is further strengthened.
5. More mainstream solvers should be included, such as [1-7]. They are a set of representing (but not limited to) neural works for routing problem solving, including supervised-, reinforced-, unsupervised-, meta-reinforced-, divide-and-conquer-, and neural-heuristic-mannered approaches. It is acceptable the authors include a subset of them into the framework, but this would benefit the completeness for your empirical evaluation.
6. The authors are also suggested to evaluate their framework on the conventionally used uniform TSP dataset (like those consistent test files through [1,2,4,5,7, etc.]). And please report the origianal objective for the COPs in addition to currently only the gap.
7. The claim in the title is broader than what is done within the paper. If the framework is to be a neural solver selection of combinatorial optimization, can it be readily applied to more complex problems beyond TSP and CVRP? And what is the solution at larger-scaled (e.g., $N\ge 1000$) instances where most neural solvers struggle to produce satisfactory results compared to the traditional heuristics?

**References:**

[1] DIMES: A Differentiable Meta Solver for Combinatorial Optimization Problems.

[2] An Efficient Graph Convolutional Network Technique for the Travelling Salesman Problem.

[3] Unsupervised Learning for Solving the Travelling Salesman Problem.

[4] Graph Neural Network Guided Local Search for the Travelling Salesperson Problem.

[5] Generalize a Small Pre-trained Model to Arbitrarily Large TSP Instances.

[6] GLOP: Learning Global Partition and Local Construction for Solving Large-scale Routing Problems in Real-time.

[7] Attention, Learn to Solve Routing Problems!

**Questions:**

Please see the weaknesses part for questions and suggestions.

---

> ### Author Response · Authors · 2024-11-22
>
> Thank you for reviewing our paper. We sincerely appreciate your valuable suggestions for refining our work. According to your comments, we have clarified our contributions and enriched our paper as you advised. Here are the detailed responses.
>
> **Response to Weakness1: The novelty and position of this paper**
>
> Thanks for your valuable comments. There may be some misunderstanding regarding the intended position and focus of this paper, and we are grateful for the chance to elaborate. The primary contribution of our work lies in pioneering the integration of model selection into Neural Combinatorial Optimization (NCO) and demonstrating its effectiveness through extensive experiments.
>
> Unlike traditional methods, NCO methods leverage neural networks to build data-driven solvers, obtaining good optimality gaps with significantly superior inference efficiency. However, inspired by the No-Free-Lunch theorem, we investigated the instance-level performance of prevailing NCO solvers and found that they demonstrate clear complementarity. This phenomenon emphasizes the potential of combining the advantages of state-of-the-art neural solvers and motivates our proposal of adaptively selecting suitable solvers for each instance. Since our work is supposed to be a pioneer attempt at neural solver selection, our main goal is to verify the possibility and benefits of solver selection for NCO. In our experiments, we found that even a straightforward method using hand-crafted features and classification models can outperform the state-of-the-art neural solver, which strongly indicates that solver selection is a promising direction for NCO. We believe our work can benefit the NCO community and inspire future research in this area. For example, we have disscussed several future directions (e.g., designing better feature representation of neural solvers, exploring runtime-aware selection methods for neural solvers with different search budgets, and enhancing the solver pool by training) in Section 5.
>
> We hope the above clarification has made the main contribution of this work clear. In fact, for the current method, there are also some meaningful technical advancements. For example,
>
> 1. **The pooling-based hierarchical encoder.** We propose to design a graph pooling method to downsample representative nodes from the complete instance, which constructs hierarchical representations that are empirically proven robust when the problem distribution/scale shifts. To the best of our knowledge, this approach is new in NCO.
> 2. **The selection strategies.** Existing selection methods for optimization algorithms or machine learning models only focus on top-1 or top-k selection, while we propose two new selection strategies (i.e., rejection-based and top-p selection) by considering the confidence of the selection model. These strategies can adaptively select additional neural solvers for low-confidence instances, enhancing the robustness with minimal time consumption.
>
> Thank you again for your thoughtful comments. We sincerely hope our response clarifies our contributions and the potential impact of our work.
>
> **Weakness2: The necessity of neural solver selection**
>
> Thanks for your comment. As you noted, sequentially performing solvers on a target dataset can be effective. However, when new instances come, this approach requires rerunning all solvers on each instance. In contrast, a well-trained selection model can generalize to unseen instances in a zero-shot manner, efficiently selecting the most suitable solver for each instance. Thus, the selection model only needs to run the selected solver, and can be much more efficient than simple sequential execution (requiring running all solvers). We hope this addresses your concerns.
>
> **Weakness3: The ambiguity of “OPT”**
>
> We are very sorry for the confusion. The results reported in our paper are just the optimality gap with respect to the output of HGS on CVRP and LKH3 on TSP. We used "OPT" to represent the performance of the best individual solver on each instance. To avoid misunderstanding, we have replaced "OPT" with "Oracle" in the revised version.
>
> **Weakness4: Discussion on the combination with traditional solvers**
>
> Thank you for your constructive suggestion. We agree that combining powerful traditional solvers and neural solvers for enhanced performance is of significance to both communities. However, a key motivation of neural combinatorial optimization is that neural networks have an overwhelming speed advantage over heuristc methods, which can serve as alternatives to traditional solvers in scenarios that require time efficiency. In this paper, we follow this motivation and aim to improve the performance of neural solvers without sacrificing their efficiency. Thus, we did not involve traditional heuristc solvers in this work.

---

> ### Author Response · Authors · 2024-11-22
>
> **Weakness5 & 7-2 New neural solver & larger-scaled experiments**
>
> Thank you for your constructive suggestions. We have added several new solvers to our pool as recommended. Then, we increased the problem scale from N ∈ [0,500]  to N ∈ [500, 2000] and used the enhanced solver pool to conduct new experiments. The results, shown in the following table, demonstrate that our framework is compatible with more neural solvers and can also improve performance over the single best solver on larger-scale instances. We hope these additional results can address your concerns.
>
> For details, we add two divide-and-conquer solvers, GLOP and UDC [8], to our solver pool, which can significantly enhance the overall performance. We do not include other neural solvers since they either contribute little to the overall performance [1,2,7], which would be filtered out by our elimination process detailed in Appendix A.3, or rely on post-search techniques (e.g., Monte-Carlo tree search) [3, 5] that consume much more time than other greedy decoding methods in the solver pool, causing some fairness issues. The construction of our solver pool now considers reinforced (ELG, INViT), supervised (BQ, LEHD), meta-learning-based (Omni), diffusion-based (DIFUSCO, T2T), and divide-and-conquer (GLOP, UDC) methods. The experimental results have shown that our proposed framework can effectively combine the advantages of these neural solvers and significantly improve performance. We have revised to include these new results (i.e., Table 12) in the new version.
>
> | Methods \ Metrics | Optimality gap on TSP500-2000 | Time on TSP500-2000 |
> | --- | --- | --- |
> | Single best solver | 6.104% | 8.369s |
> | Ours (Greedy) | 5.540% (0.038%) | 8.322s (0.036s) |
> | Ours (Top-k, k=2) | **5.369% (0.003%)** | 15.566s (0.085s) |
> | Single best of new solver pool | 3.562%  | 5.274s  |
> | Ours with new solvers (Greedy) | 3.126% (0.002%) | 6.892s (0.006s) |
> | Ours with new solvers (Top-k, k=2) | **2.955% (0.005%)** | 13.713s (0.036s) |
>
> **Weakness6: Evaluation on uniform TSP dataset**
>
> Thanks for your thoughtful suggestions. Yes, uniform datasets are commonly used, and many neural solvers have already achieved excellent performance (Gap < 0.5%) on the uniform TSP100. Our study focuses on a harder setting by using a dataset with diverse instances of varying distributions and scales, which allows us to assess whether a selection method can effectively identify the suitable solver for a wide range of instances. Thanks to your suggestion, we also provide the results of coordinating multiple neural solvers on the uniform TSP100, as shown in the following table. These results show that while selection on this dataset can still be effective, the potential improvement over the best single solver is limited, as single solvers already perform well on the uniform dataset.
>
> | Methods | Our synthetic dataset | Uniform TSP100 |
> | --- | --- | --- |
> | Single best solver | 2.33% | 0.29% |
> | Oracle of multiple solvers | 1.24% | 0.10% |
>
> We have revised to add these results (i.e., Table 14) in the new version. Regarding your second suggestion, we have also revised to report the original objective values (i.e., Tables 8 and 9).
>
> **Weakness7-1 Versatility of our framework**
>
> Thanks for your valuable question. In this paper, we implemented the method on TSP and CVRP since these two representative problems are widely studied in the NCO community and have many diverse neural solvers for selection. Besides TSP and CVRP, our proposed selection framework is adaptable to other problems. For new problems, one only needs to customize the feature extraction component. For instance, when adapting our framework to scheduling problems, one can adjust the graph attention encoder according to MatNet [9] (i.e., add edge embeddings). Other components, like training loss and selection strategies, do not require changes for new problems. We have revised to add some discussion in Section 5. Thank you very much.
>
> References:
>
> [1-7] correspond to the references you provided
>
> [8] UDC: A unified neural divide-and-conquer framework for large-scale combinatorial optimization problems. In Advances in Neural Information Processing Systems 37 (NeurIPS).
>
> [9] Matrix encoding networks for neural combinatorial optimization. In Advances in Neural Information Processing Systems 34 (NeurIPS).

---

### Official Review · Reviewer_PbuG · 2024-11-03

**Soundness:** 2
**Presentation:** 3
**Contribution:** 2
**Rating:** 6
**Confidence:** 4

**Summary:**

This work proposes a neural solver selection framework to efficiently select a subset of suitable neural combinatorial optimization (NCO) solvers to handle each problem instance at the inference time. Three key components (feature extraction, training loss, and selection strategies) have been proposed and investigated in detail for solver selection. Experimental results show that the proposed framework can achieve promising performance on the traveling salesman problem (TSP) and the capacitated vehicle routing problem (CVRP) with little extra time cost.

**Strengths:**

+ This paper is well written and easy to follow.

+ Algorithm selection is an important strategy for classic (combinatorial) optimization, and it has not yet been well studied for NCO. This work is a timely contribution to this important research direction.

+ The proposed algorithm selection framework can achieve promising performance on different TSP and CVRP instances.

**Weaknesses:**

**1. Connection to Neural Combinatorial Optimization (NCO) and Novelty**

Although this work's main motivation is to propose a neural solver selection framework for NCO, it seems that the proposed solver selection approach is actually agnostic to NCO. It is more like an independent learning-based solver selection method that can be used for other solvers, including the traditional ones. What makes the proposed method specific for NCO?

On the other hand, algorithm selection is already a popular research direction in the optimization community. As correctly mentioned in this paper, many (learning-based) algorithm/solver selection methods have already been proposed and widely used in practice (for example, see [1] for TSP algorithm selection). Many of them can be easily adapted to select NCO solvers. What is the novelty/contribution of the proposed framework over the existing algorithm selection methods?

[1] https://tspalgsel.github.io/

**2. Discussion/Comparison with Existing Algorithm Selection Methods**

I think the claim "[the traditional method] has never been explored in the area of neural combinatorial optimization" is far from enough to truly distinguish the proposed method from the traditional algorithm selection method. A detailed discussion/comparison with traditional algorithm selection methods is needed.

What are the advantages/disadvantages of the proposed method compared with existing (learning-based and non-learning-based) algorithm selection methods? What is the performance of existing algorithm selection methods with NCO solvers? What is the performance of the proposed method with classic solvers?

**3. Novelty of the Key Components**

The proposed framework has three key components, namely feature extraction, selection model, and selection strategies. However, it seems that these components and the proposed structures are quite common in the NCO and algorithm selection community. The novelty and unique contribution of these components should be highlighted with solid evidence.

**4. Generalization Performance**

Although experimental results show the proposed framework has good performance with problem distribution/scale shifts, it is unclear why it can achieve good out-of-distribution generalization performance as a learning-based method.

**5. Experiments**

In the experiments, only a single summary table is provided for each comparison. I think a complete table with separate results for different instances (e.g., with different numbers of nodes), as widely used in other NCO papers, could be very helpful to better understand the performance of the proposed method.

As mentioned above, a detailed comparison with existing (learning-based and non-learning-based) algorithm selection methods is also needed.

**Questions:**

See weaknesses.

---

> ### Author Response · Authors · 2024-11-22
>
> Thank you for reviewing our paper. We sincerely appreciate your valuable comments, which are very helpful in refining our work. However, there may exist some misunderstanding about the position we expected of this paper. We have carefully revised our paper according to your comments and try to clarify our motivation and contribution. Here are the detailed responses.
>
> **Response to “Connection to Neural Combinatorial Optimization and Novelty”**
>
> Thank you for your valuable questions. We first want to emphasize that our main contribution is introducing algorithm selection to the community of Neural Combinatorial Optimization (NCO) for the first time, and showing its effectiveness even with a very straightforward implementation. Unlike traditional methods, NCO methods leverage neural networks to build data-driven solvers, obtaining good optimality gaps with significantly superior inference efficiency. However, inspired by the No-Free-Lunch theorem, we investigated the instance-level performance of prevailing NCO solvers and found that they demonstrate clear complementarity. This phenomenon emphasizes the potential of combining the advantages of state-of-the-art neural solvers and motivates our proposal of adaptively selecting suitable solvers for each instance. Since our work is supposed to be a pioneer attempt at neural solver selection, our main goal is to verify the possibility and benefits of solver selection for NCO. In our experiments, we found that even a straightforward method using hand-crafted features and classification models can outperform the state-of-the-art neural solver, which strongly indicates that solver selection is a promising direction for NCO. We believe our work can benefit the NCO community and inspire future research in this area.
>
> On the other hand, the implementation of our neural solver selection framework has some advanced components. For example, we propose a new method of extracting instance features for NCO, which is different from previous works on the classical algorithm selection for TSP. Firstly, we verified the manual features proposed in classical algorithm selection for TSP [1] and found that they can only achieve limited performance (in Table 3 of the original paper). To address this, we proposed a novel pooling-based hierarchical encoder designed to extract richer instance features, leading to significantly better generalization performance. We believe that such an instance feature extraction method may also be helpful for improving other NCO methods, not limited to our neural solver selection framework.
>
> Thank you again for your thoughtful comments. We sincerely hope the above clarification has made the main contribution of this work clear.
>
> **Response to “Comparison with existing selection methods”**
>
> Thank you for your valuable question. As we mentioned above, the primary contribution of our work lies in pioneering the integration of model selection into NCO. Thus, we mainly gave a straightforward implementation of the selection framework, and demonstrated its superiority against the best single individual solver through extensive experiments. This has achieved the goal of this work. But we agree that it is meaningful to add the discussion and comparison with existing selection methods from other areas.
>
> Thanks to your suggestion, we have revised to discuss the difference with selection methods of non-neural solvers for TSP, and conduct detailed comparison experiments (as shown in the following table). Most algorithm selection methods follow a pipeline with two steps: Feature extraction and selection model training. Below, we summarize how our approach improves upon these steps:
> 1. For the feature extraction step, existing works for TSP [1,2,4] typically rely on hand-crafted features derived from cluster analysis, nearest-neighbor graphs, and other techniques. In contrast, we proposed a hierarchical graph encoder to learn feature representations in a data-driven manner. Both the original experiments (Table 3) and the newly added results demonstrate that our neural encoder significantly outperforms hand-crafted features, particularly in terms of generalization ability on unseen datasets.
> 2. For the selection model step, to our knowledge, most works [1,2,4] for TSP utilize traditional classification models like random forests or support vector machines, while we use a neural network trained by a learn-to-rank loss. We believe our neural selection model is a better choice so we didn't compare it with traditional models in our original experiments. Thanks to your suggestion, we include ablation of the neural selection model in the newly added experiments, which show that the neural selection model is better than the random forest.
>
> (Limited by space, the following contents are in the next block)

---

> ### Author Response · Authors · 2024-11-22
>
> (Following the block above...)
>
> Moreover, we also proposed adaptive selection strategies considering the confidence of the selection model, which goes beyond what traditional methods offer. Together, our neural encoder, learn-to-rank selection model, and adaptive strategies form a practical framework for neural solver selection, as validated by both the original and newly added experiments.
>
> For details of the newly added experiments, we introduce them as follows. To demonstrate the effectiveness of our proposed techniques, **we provide additional comparisons between our proposed method and existing algorithm selection methods for non-neural TSP solvers [1,2]**. In fact, the method of using features from [1] and our ranking model was also compared in Table 3 of the original paper. The R package *salesperson*[3] provides the up-to-now most comprehensive collection of features for TSP and is widely used in algorithm selection methods [2,4]. Based on the feature set of *salesperson*, we reproduce an advanced algorithm selection method [2] following the pipeline that computes hand-crafted features, conducts feature selection, and applies random forest for classification, where we employ univariate statistical tests to select important features. Besides, we also combine the *salesperson* features with our ranking model for ablation.
>
> | Methods \ Metrics | Optimality gap on synthetic TSP | Time on synthetic TSP | Optimality gap on TSPLIB | Time on TSPLIB |
> | --- | --- | --- | --- | --- |
> | Single best solver | 2.33% | 1.45s | 1.95% | 1.74s |
> | Features from [1] + Ranking | 1.97% (0.01%) | 1.37s (0.01s) | 1.83% (0.03%) | 1.32s (0.05s) |
> | Method of [2] | 2.12% (0.04%) | 1.35s (0.00s) | 1.56% (0.01%) | 1.34s (0.05s) |
> | Features from *salesperson* + Ranking | 1.95% (0.01%) | 1.33s (0.03s) | 1.55% (0.03%) | 1.27s (0.06s) |
> | Ours + Greedy | 1.86% (0.01%) | 1.33s (0.01s) | 1.33% (0.06%) | 1.28s (0.03s) |
> | Ours + Top-p (p=0.5) | **1.68% (0.02%)** | 1.86s (0.07s) | 1.28% (0.04%) | 1.46s (0.06s) |
> | Ours + Rejection (20%) | 1.75% (0.02%) | 1.63s (0.01s) | **1.26% (0.03%)** | 1.51s (0.04s) |
>
> The experimental results in the above table show that our proposed method can achieve superior performance than advanced algorithm selection methods on both synthetic TSP and TSPLIB. Comparing the fifth and sixth rows, our proposed hierarchical encoder demonstrates superior performance over the *salesperson* features, especially on the out-of-distribution benchmark TSPLIB. Additionally, the comparison of the fourth and fifth rows shows that our deep learning-based ranking model achieves better results than traditional classification methods. Furthermore, the results of the last three rows illustrate that our proposed adaptive selection strategies effectively enhance optimality with minimal increases in time consumption.
>
> In summary, though our goal is to introduce model selection into the area of NCO and show its effectivenss, the proposed implementaion of the selection framework for this purpose also has some technical novelty over the existing algorithm selection methods from other areas. We hope our discussion, along with the newly added experiments, addresses your concerns. Thank you again for your thoughtful feedback.
>
> **Response to “Novelty of the key components”**
>
> Thank you for your valuable question. As we emphasized before, our goal is to introduce model selection into the area of NCO and show its effectivenss. Thus, we focus on providing a practical implementation of the selection framework, instead of developing entirely new components. But the proposed implementaion of the selection framework also has some technical novelty, as we introduced before. Here, we give more detailed introduction.
>
> 1. **Pooling-Based Hierarchical Graph Encoder**: The proposed hierarchical encoder employs a graph pooling operation to downsample representative subgraphs, constructing hierarchical representations that are robust to distributional shifts, which is not quite common in the NCO community. This encoder outperforms the standard graph encoder and hand-crafted features, especially on out-of-distribution datasets. The corresponding results can be found in Table 3, and we also provide additional results (in the above table) by comparing our encoder with more hand-crafted features [1,2] according to your suggestions.
> 2. **Adaptive Selection Strategies**: We propose new adaptive strategies like top-p and rejection-based selection, which allow the model to adaptively choose one or multiple solvers based on its confidence, effectively balancing optimality and efficiency. The corresponding results can be found in Tables 1 and 2. To our knowledge, existing selection methods usually focus on the top-1 or top-k selection and have not explored such adaptive strategies.
>
> (Limited by space, the following contents are in the next block)

---

> ### Author Response · Authors · 2024-11-22
>
> (Following the block above...)
>
> 3. **Neural Solver Feature**: We explore the usage of the neural solver feature in Section 5, and its experimental results can be found in Appendix A.9. Specifically, we propose to use a Transformer to learn a summary feature from representative instances of each neural solver. This enables generalization to unseen solvers, allowing us to add new solvers to the pool without fine-tuning the selection model. To our knowledge, the generalization ability over solvers is also new for existing algorithm/model selection methods.
>
> We believe that many future efforts on selection techniques (e.g., designing better feature representation of neural solvers, exploring runtime-aware selection methods for neural solvers with different search budgets, and enhancing the solver pool by training) can be made under our framework, as discussed in Section 5. We believe our work can benefit the NCO community and inspire future research in this area.
>
> References:
>
> [1] Understanding TSP difficulty by learning from evolved instances. In LION.
>
> [2] Deep Learning as a Competitive Feature-Free Approach for Automated Algorithm Selection on the Traveling Salesperson Problem. In PPSN.
>
> [3] Salesperson: Computation of instance features and R interface to the state-of-the-art exact and inexact solvers for the traveling salesperson problem. [https://github.com/jakobbossek/salesperson](https://github.com/jakobbossek/salesperson.git).
>
> [4] On the Potential of Normalized TSP Features for Automated Algorithm Selection. In FOGA.
>
> **Response to: Doubts on the generalization performance**
>
> Enabling neural solvers for combinatorial optimization to generalize across scales and distributions is a prevailing topic in the community of NCO[4,5], which is important to improve its performance in practice. One supporting reason to realize this is that TSP/CVRP instances with different scales and distributions potentially share similar characteristics [6,7]. For example, even for instances from different distributions, there may exist similar local patterns (i.e., local sub-areas), making their intrinsic characteristics consistent. Our proposed hierarchical encoder can effectively capture such local patterns using graph pooling (e.g., intuitive illustrations in Figure 6), enabling reasonable out-of-distribution ability. To examine the generalization performance of our proposed instance-level neural solver selection framework, we do experiments in scenarios with distribution/scale shifts. The results in Table 2 demonstrate the effectiveness of our method.
>
> References:
>
> [4] Learning the travelling salesperson problem requires rethinking generalization. In Constraints.
>
> [5] Towards omni-generalizable neural methods for vehicle routing problems. In ICML.
>
> [6] INViT: A generalizable routing problem solver with invariant nested view transformer. In ICML.
>
> [7] Towards generalizable neural solvers for vehicle routing problems via ensemble with transferrable local policy. In IJCAI.
>
> **Response to: Comparison under different scales of instances**
>
> Thank you for your thoughtful comments. As you suggested, we have revised to provide separate results on our datasets for a deeper investigation. The following tables demonstrate that our selection method consistently outperforms the single best solver across different problem scales on both TSP and CVRP datasets. We hope this addresses your concerns.
>
> Separate results according to problem scale $N$ on the synthetic TSP dataset. We report the mean (standard deviation) optimality gap over five independent runs.
>
> | Methods | $50\le N \le 200$ | $200< N \le 300$ | $300< N \le 400$ | $400< N \le 500$ |
> | --- | --- | --- | --- | --- |
> | Single best solver | 0.96% | 2.34% | 2.78% | 2.98% |
> | Oracle | 0.39% | 1.19% | 1.70% | 2.18% |
> | Ours (Greedy) | 0.84% (0.03%) | 2.01% (0.02%) | 2.43% (0.02%) | 2.71% (0.03%) |
> | Ours (Top-k, k=2) | 0.61% (0.02%) | 1.53% (0.03%) | 1.99% (0.03%) | 2.41% (0.05%) |
> | Ours (Rejection, 20%) | 0.75% (0.04%) | 1.86% (0.04%) | 2.33% (0.03%) | 2.62% (0.02%) |
> | Ours (Top-p, p=0.5) | 0.71% (0.02%) | 1.70% (0.02%) | 2.24% (0.04%) | 2.57% (0.04%) |
>
> Separate results according to problem scale $N$ on the synthetic CVRP dataset.
>
> | Methods | $50\le N \le 200$ | $200< N \le 300$ | $300< N \le 400$ | $400< N \le 500$ |
> | --- | --- | --- | --- | --- |
> | Single best solver | 3.95% | 6.06% | 7.76% | 9.24% |
> | Oracle | 2.17% | 4.33% | 5.74% | 7.40% |
> | Ours (Greedy) | 2.85% (0.03%) | 4.87% (0.02%) | 6.47% (0.05%) | 8.09% (0.01%) |
> | Ours (Top-k, k=2) | 2.32% (0.02%) | 4.54% (0.02%) | 5.91% (0.03%) | 7.55% (0.03%) |
> | Ours (Rejection, 20%) | 2.64% (0.02%) | 4.70% (0.03%) | 6.22% (0.02%) | 7.91% (0.03%) |
> | Ours (Top-p, p=0.8) | 2.36% (0.02%) | 4.70% (0.04%) | 6.21% (0.05%) | 7.81% (0.02%) |

---

> ### Comment · Reviewer_PbuG · 2024-11-27
>
> Thank you very much for your thorough response and new experimental results. I have also read other reviewers' comments and the corresponding responses. Since many of my concerns have been properly addressed, I raise my score to 6.
>
> The major remaining concern is still on its connection to NCO. The main contribution of this work is more like 1) a new learning-based solver selection approach that can be used for different (neural or traditional) solvers and 2) a case study of using the proposed method for NCO, rather than a novel approach that can truly leverage the specific patterns/characteristic of NCO models for neural solver selection. The potential research directions briefly discussed in the conclusion section (feature extraction for neural solvers, runtime-aware selection, and actively training complementary solvers) are all important and could be very helpful in achieving this goal, but none of them have been done in this work.
>
> On the other hand, I agree with the authors that solver selection is important for NCO, and this work might inspire more follow-up works on this research direction. Therefore, I vote to weakly accept this work (6).

---

> > ### Author Response · Authors · 2024-11-27
> > **Thanks for your feedback**
> >
> > Thank you very much for your kind response. We are very pleased to hear that our reply has addressed many of your concerns. Yes, we truly hope this work can open a door for solver selection of NCO, and inspire more follow-up works on this research direction. Thank you once again for your time and valuable insights in reviewing our paper.

---

### Official Review · Reviewer_kQip · 2024-11-03

**Soundness:** 2
**Presentation:** 3
**Contribution:** 3
**Rating:** 6
**Confidence:** 3

**Summary:**

This paper proposes a novel framework for selecting the most suitable neural solver for different instances of combinatorial optimization problems (COPs). The framework effectively combines graph feature extraction through attention mechanisms and hierarchical encoders, as well as multiple solver selection strategies, including Greedy, Top-k, and Rejection-based approaches. The experimental results demonstrate the superiority of the proposed method over traditional approaches on tasks like TSP and CVRP. Overall, the paper offers valuable insights to instance-specific solver selection.

**Strengths:**

1. The paper introduces a novel combination of hierarchical graph encoders and multiple selection strategies, which together enhance solver performance for different combinatorial optimization problem instances.
2. The experimental results cover a range of combinatorial optimization tasks and demonstrate improvements in performance compared to using a single solver.
3. The proposed Adaptive Solver Selection Framework for selecting solvers based on instance characteristics is flexible.

**Weaknesses:**

1. The current selection strategies include Greedy selection, Top-k, Rejection-based, and Top-p. While these strategies have demonstrated effectiveness in different experimental settings, the basis for choosing the most suitable strategy for different types of instances is not clear. For example, what kind of instances would make Top-k more suitable than Top-p?
2. The paper could benefit from including more graphical representations.

**Questions:**

1. The paper mentions that the hierarchical encoder can better leverage the hierarchical structural features in COPs. However, the intuitive interpretation of these hierarchical features is unclear. How do these structures correspond to specific instance properties of problems such as TSP or CVRP?
2. The paper mentions the use of graph features and instance scale as inputs for the selection model, while the specific features of different neural solvers are not directly involved in the learning process. Would the absence of these solver-specific features limit the generalization ability of the selection model?
3. During the score calculation phase, how exactly does the MLP relate to different solvers? In other words, how are the features of different solvers reflected in the MLP, and how does this ensure that the classification results are correlated with the solvers' features?
4. In the Top-p selection strategy, the paper defines a threshold probability $p$ to decide which solvers to retain. Is this threshold set adaptively based on the problem's features, allowing for optimal performance?
5. The introduction of a hierarchical encoder adds complexity to the model. How does this impact the overall training efficiency and inference speed of the model? Is there any quantitative analysis showing the trade-off between the hierarchical encoder's added complexity and the model's performance improvements?

---

> ### Author Response · Authors · 2024-11-22
>
> Thank you for your valuable comments. We sincerely appreciate your agreement on the effectiveness of the instance-specific solver selection proposed in our paper, which, we believe, has the potential to be a new branch of techniques for the application of NCO solvers. Meanwhile, we are very sorry for the possible unclear description, which may lead to confusion. We have carefully taken your reviews into account and revised our paper. Here are our detailed responses to your comments and questions, which we hope will address your concerns.
>
> **Response to weakness 1: Lack of discussion on different selection strategies**
>
> Thank you for your comments. We have revised our paper to clarify the advantages of each selection strategy. According to the mechanisms of the four selection strategies, they have different preferences in the trade-off of efficiency and optimality. Generally, for efficiency, Greedy > Rejection ≈ Top-p > Top-k; for optimality, Top-k > Rejection ≈ Top-p > Greedy. Meanwhile, their hyper-parameters can be used for balancing efficiency and optimality as well. As a result, the choice of selection strategies can be decided by the users according to their preference, and we suggest using Top-p or Rejection as the default choice since they can adaptively select solvers based on the confidence of the selection model.
>
> **Response to Question 1: Detailed interpretation of the hierarchical encoder**
>
> Feature extraction plays an important role in selecting solvers for each instance. To obtain better instance features, we propose the hierarchical encoder which combines multi-level features together. In TSP and CVRP, certain nodes, such as those in clusters or geometric patterns, are particularly representative and informative for describing instance properties. By focusing on these nodes, we can effectively capture the spatial distribution of the entire instance. Our proposed hierarchical encoder is designed to identify these key nodes and create downsampled graphs, allowing it to concentrate on representative subgraphs and learn more robust features. The detailed processes are described in Section 3.1 of our main paper.
>
> The overall performance of the hierarchical encoder, as shown in Table 3 of the main paper, demonstrates its superiority, especially across scales and distribution scenarios (TSPLib and CVPRLib). Thanks to your suggestion, we have revised to add Figure 6 to illustrate the retained nodes after downsampling. We can find some consistent patterns which are intuitively reasonable. We summarize them as three main points:
>
> 1. Cluster nodes. As illustrated in Figures 6(a) and 6(b), when instances contain certain clusters, the hierarchical encoder tends to select a subset of “representative” nodes from each cluster, efficiently describing the entire spatial distribution.
> 2. Specific blocks. As illustrated in Figures 6(c) and 6(d), when instances contain specific complex geometric patterns like squares (Figure 6(c)) and arrays (Figure 6(d)), the hierarchical encoder can capture the nodes of these important areas to identify its characteristics.
> 3. Boundary nodes. For instances without clear sub-components, the hierarchical encoder tends to focus on boundary nodes that describe the global shape, as illustrated in Figures 6(e) and 6(f).
>
> For your convenience, we also put the figures at [https://anonymous.4open.science/r/pics_for_analyzing_encoder-CA27/illustration_of_nodes.pdf](https://anonymous.4open.science/r/pics_for_analyzing_encoder-CA27/illustration_of_nodes.pdf).
>
> Thank you very much for your valuable suggestion, which really has improved our work.
>
> **Response to Questions 2 & 3 about neural solver features**
>
> Thanks for your thoughtful questions. When solver-specific features are absent, we associate neural solvers with the index of the MLP output. For instance, the first dimension of the MLP output corresponds to the score for the first neural solver. Extensive experiments in our paper have demonstrated that this simple approach has achieved good generalization across instances.
>
> Under our proposed selection framework, constructing features of neural solvers and utilizing them for selection is very promising but challenging. In this paper, we made a preliminary exploration for integrating solver-specific features into the learning process, detailed in Section 5 and Appendix A.9. Our preliminary method involves learning a summary feature from representative instances of each neural solver. This method facilitates generalization to new neural solvers, allowing us to add them to the solver pool without fine-tuning the selection model. However, we did not observe an improvement in generalization over instances using this method. This suggests that further research is required to develop more sophisticated neural solver features, which could enhance model capacity and generalization performance. We will further study it in our future work.

---

> ### Author Response · Authors · 2024-11-22
>
> **Response to Question 4: The threshold hyperparameter in Top-p selection strategy**
>
> Thank you for your insightful question. Currently, the threshold parameter $p$ is not adaptively set for each instance but fixed (e.g., 0.5 on TSP) for all the instances. In Figure 4 of our paper, we plotted the results of using $p$ values ranging from 0.95 to 0.40 in decrements of 0.01. These results demonstrate a trade-off: As $p$ increases, the optimal gap improves, but the average time required also increases. This highlights $p$ as a hyperparameter that allows users to balance efficiency and optimality. Though using a fixed value has led to good performance in our experiments, we believe that adaptively adjusting $p$ for each instance could further improve performance as you suggested. We have revised to add this as an interesting direction for future research.
>
> **Response to Question 5: Efficiency of the proposed hierarchical encoder**
>
> Thanks for your question. The introduction of our hierarchical encoder brings very limited computation costs. To address your concerns, we provide detailed comparisons of the computation cost and optimality, between our hierarchical encoder and a typical graph encoder. The results are shown in the following table, which includes the inference time per instance on TSPLIB, training time per epoch, and the average optimality gap on TSPLIB.
>
> | Methods | Inference time on TSPLIB of selection model | Inference time on TSPLIB of neural solvers | Training time each epoch | Optimality gap on TSPLIB |
> | --- | --- | --- | --- | --- |
> | Naive graph encoder | 0.0054s | 1.2600s | 1m40s | 1.54% |
> | Hierarchical graph encoder | 0.0070s | 1.2961s | 2m30s | 1.37% |
>
> We can observe from the second column that the introduction of our hierarchical encoder will increase the inference time of the selection model a little bit, e.g., from 0.0054s to 0.0070s. However, as shown in the second and third columns, the inference time of the selection model is orders of magnitude shorter than that of the neural solvers, so the inference efficiency of the selection model is less of a concern. The fourth column shows that the training time per epoch of the naïve encoder and the hierarchical encoder are 1m40s and 2m30s, respectively. Although the hierarchical encoder slows the training, the total runtime for 50 epochs is still only 2 hours, which is acceptable in most scenarios. Therefore, the performance metric (i.e., optimality gap) of different encoders is more crucial, especially the generalization performance. If the encoder learns robust representations, we can directly transfer the selection model to different datasets in a zero-shot manner, saving the time for fine-tuning and adaptation. Considering the better generalization (e.g., the optimality gap decreases from 1.54% to 1.37%), we believe that the proposed hierarchical encoder is a better choice.
>
> Thanks to your suggestion, we have revised to add the results (i.e., Table 6) in the new version. We hope this explanation can address your concerns. Thank you again.

---

> > ### Comment · Reviewer_kQip · 2024-12-03
> >
> > I appreciate the authors' response, which addressed many of my concerns. However, I still have the following reservations: Although the paper presents a general framework for neural solver selection, it lacks a breakthrough in method design compared to existing work. Many of the proposed selection strategies, such as Top-k, rejection-based, and Top-p selection, are direct applications of existing ensemble learning methods without introducing mechanisms that could substantially enhance selection efficiency or effectiveness. While these strategies do provide some improvement, I remain concerned that the contributions may not be sufficiently innovative to meet the standards of this conference. However, after considering all factors, I am willing to raise my score to 6.

---

> > > ### Author Response · Authors · 2024-12-04
> > > **Thanks for your feedback**
> > >
> > > Thank you very much for your feedback. We are very pleased to hear that our reply has addressed many of your concerns.
> > >
> > > The aim of this work is introducing the idea of neural solver selection to the NCO community for the first time, and showing its effectiveness with a straightforward implementation. Thus, we did not focus on the design of the components in our proposed selection framework. We fully agree that designing better components is interesting, e.g., designing better feature representation of neural solvers, exploring runtime-aware selection methods for neural solvers with different search budgets, and enhancing the solver pool by training, as we discussed in Section 5. We believe our work can open a new line for NCO, and inspire more follow-up works on neural solver selection.
> > >
> > > Thank you again for dedicating your time and effort to review our paper.

---

### Official Review · Reviewer_N15X · 2024-11-03

**Soundness:** 3
**Presentation:** 3
**Contribution:** 3
**Rating:** 6
**Confidence:** 4

**Summary:**

This paper considers a new perspective on solving the Combinatorial Optimization (CO) problem using deep learning. Given an instance, a deep learning framework is trained to select the best suitable solver for this instance from a state-of-the-art solver pool. The general idea is (1) feature extraction of the input instance, (2) selection based on several criteria, e.g., top k, and the output of a trained classifier/ranking model.  (3) run the instance on the select solver(s).

The experiment results show that the proposed method can improve the current single solver performance with few efforts. It also shows the ability to generalize. The key idea behind this paper is similar to this paper: Bai Y, Zhao W, Gomes C P. Zero Training Overhead Portfolios for Learning to Solve Combinatorial Problems[J]. arXiv preprint arXiv:2102.03002, 2021. Since the CO problems are typically too hard, so a single sovler cannot capture the entire problem structure. So, different solvers have their own advantages, then we can leverage this to improve the performance.

**Strengths:**

(1) Since the CO problems are typically too hard, a single solver cannot capture the entire problem structure. Different solvers have their own advantages, which we can leverage to improve performance.
(2) The experiment results show the ability to generalize.

**Weaknesses:**

See questions.

**Questions:**

(1) Do you have any results on TSP-10000 or large instances? Trying to train your selection model on TSP-1000 and see how it can be generalized to TSP-10000 is critical.
(2) Are the instances training the selection model generated from the same distribution of the testing instances?

---

> ### Author Response · Authors · 2024-11-22
>
> Thank you for your positive review. Here are our detailed responses to your comments and questions, which we hope will address your concerns.
>
> **Response to your comment: The key idea behind this paper is similar to the paper: Zero Training Overhead Portfolios for Learning to Solve Combinatorial Problems (ZTop).**
>
> Thanks for pointing out this related work. The ZTop method uses a fixed set of neural solvers to construct a portfolio for all instances, similar to other ensemble and population-based methods [1,2], which we discussed in the paper. Instead of employing a static portfolio of solvers, our method provides a more flexible solution by adaptively creating instance-specific portfolios, i.e., adaptively selecting the most suitable solvers for each instance. In fact, our experiments in Figure 3 have demonstrated that our proposed top-k strategy consistently outperforms the static portfolios (similar to ZTop) across different k values. We have revised to add some discussion about the relationship between our method and ZTop. Thank you.
>
> **Question 1: Results on large-scale instances like TSP-10000**
>
> Thanks for your suggestion. The generalization performance of neural solvers on large-scale instances still remains to be an important challenge in the community. As our work focuses on exploring the benefits of instance-level neural solver selection, our experiments mainly follow common settings with scales under 1000, which is friendly to the prevailing methods. Thanks to your suggestion, we additionally examine larger-scale cases with larger-scale instances and newly added divide-and-conquer solvers [3,4]. However, solving TSP-10000 is too hard and memory-consuming for the most prevailing methods, so we increase the scale to $N=2000$ for the evaluation of large-scale performance. The results show that our selection framework can generalize to larger-scale instances where $N\ge1000$. We have revised to add these results (i.e., Table 12) in the new version. We hope this can address your concerns.
>
> | Methods \ Metrics | Optimality gap on TSP500-2000 | Time on TSP500-2000 |
> | --- | --- | --- |
> | Single best solver | 6.104% | 8.369s |
> | Ours (Greedy) | 5.540% (0.038%) | 8.322s (0.036s) |
> | Ours (Top-k, k=2) | **5.369% (0.003%)** | 15.566s (0.085s) |
> | Single best of new solver pool | 3.562%  | 5.274s  |
> | Ours with new solvers (Greedy) | 3.126% (0.002%) | 6.892s (0.006s) |
> | Ours with new solvers (Top-k, k=2) | **2.955% (0.005%)** | 13.713s (0.036s) |
>
> **Question 2: Are the instances training the selection model generated from the same distribution of the testing instances?**
>
> In the experiments, we evaluate our proposed method under two test settings: 1. In-Distribution: The instances for test are sampled from the same synthetic distribution with instances for training. 2. Out-of-Distribution: Instances sampled from synthetic distribution are used for training, and the popular problem library TSPLIB and CVRPLIB are used for test. For more details, please refer to Section 4.1 of our paper.
>
> Thank you again for your valuable comments. We sincerely hope our response can answer your questions, and any further questions and discussions are very welcome.
>
> References:
>
> [1] Ensemble-based deep reinforcement learning for vehicle routing problems under distribution shift. In Advances in Neural Information Processing Systems 36 (NeurIPS).
>
> [2] Winner takes it all: Training performant RL populations for combinatorial optimization. In Advances in Neural Information Processing Systems 36 (NeurIPS).
>
> [3] GLOP: Learning global partition and local construction for solving large-scale routing problems in real-time. In Proceedings of the 38th AAAI Conference on Artificial Intelligence (AAAI).
>
> [4] UDC: A unified neural divide-and-conquer framework for large-scale combinatorial optimization problems. In Advances in Neural Information Processing Systems 37 (NeurIPS).

---

### Author Response · Authors · 2024-11-22
**General response to all reviewers**

We are very grateful to the reviewers for carefully reviewing our paper and providing constructive comments and suggestions. We have revised the paper carefully according to the comments and suggestions. The changed and newly added parts are colored in blue in the new version. Our response to individual reviewers can be found in the personal replies, but we would also like to make a brief summary of revisions for your convenience.

1. According to the suggestion of Reviewer N15x and SaPT, we conduct new experiments on larger-scale instances (N up to 2000) with newly added divide-and-conquer neural solvers, detailed in Appendix A.12. The additional results demonstrate that our proposed method can be compatible with more neural solvers and can also improve performance on larger-scale instances.
2. According to the suggestion of Reviewer PbuG, we conduct comparison experiments with advanced algorithm selection methods for non-neural TSP sovlers, showing the superiority of our implementation. Details can be found in Appendix A.13.
3. According to the suggestion of Reviewer kQip, we provide graphical illustrations of how the hierarchical encoder works in Appendix A.15, where we find some intuitively reasonable down-sampled patterns.
4. We provide more detailed results of our method.

   (1). According to the suggestion of Reviewer PbuG, we present separate results with different problem scales in Appendix A.11, demonstrating that our method consistently outperforms individual neural solvers across different problem scales.

   (2). According to the suggestion of Reviewer kQip, we add comparisons of computational costs between our hierarchical encoder and a typical graph encoder in Appendix A.6. These results further validate the efficiency and superiority of our hierarchical encoder.

   (3). According to the suggestion of Reviewer SaPT, we now include the results of objective values in Tables 8 and 9 for more detailed comparisons.
5. We add some discussions according to the reviewers' comments.

   (1). As suggested by Reviewer kQip, we have expanded the discussion on the advantages of our proposed selection strategies in Appendix A.7, highlighting their impact on optimality and efficiency;

   (2). In Appendix A.14, we have added explanations for our dataset choices as suggested by Reviewer SaPT, clarifying how the dataset aligns with the scope and goals of our work;

   (3). In Section 5, we have included a discussion on the versatility of our framework, as also recommended by Reviewer SaPT, emphasizing its potential applicability to different combinatorial optimization problems;

   (4). Following Reviewer N15X's suggestion, we discuss a new related work in Appendix A.5 to provide a more comprehensive context for our contributions.


**We hope that our response has addressed your concerns, but if we missed anything please let us know.**

---

> ### Author Response · Authors · 2024-11-25
>
> Dear Reviewers,
>
> Thank you for dedicating your time and effort to reviewing our paper. In response to your valuable comments and questions, we have made significant efforts to provide additional discussions and experimental results. As the ICLR public discussion phase will be ending in less than 2 days, we would like to kindly remind you and ask if our responses could address your concerns. Any further questions and comments are also welcomed!
>
>
>
> Best Regards,
> Authors

---

### Meta-Review · Area_Chair_U1hn · 2024-12-21

**Metareview:**

This paper proposed a learning based solver selection method for neural vehicle routing models. It involves feature extraction, selection model, and selection strategy, aiming to allocate each instance to the most suitable solver from a pool of neural VRP models. Reviewers agreed that the proposed method is interesting and neural solver selection is of practical meaning. However, they also raised several key concerns including 1) insufficient discussion and comparison to existing algorithm selection methods; 2) insufficient link to NCO; and 3) insufficient technical novelty. I particularly agree with the first point. Algorithm selection is a classic topic with many mature methods that can easily be applied to select neural VRP solvers, but this paper lacks a systematic discussion of the literature, as well as proper comparison to SOTA methods. In addition, the link to NCO is indeed not strong, as it appears to be a neural algorithm selection method that can be applied to other algorithms. So overall, this paper is interesting, but still requires major improvement to reach the acceptance threshold.

**Additional Comments On Reviewer Discussion:**

Authors provided detailed responses with additional results. However, the key concerns mentioned above still remain. Though two reviewers increased their score, the overall evaluation is still borderline.

---

### Decision · Program_Chairs · 2025-01-22

Reject